# Early warning system for ice collapses and river blockages in the Sedongpu Valley, southeastern Tibetan Plateau

Wei Yang[1]*, Zhongyan Wang[1], Baosheng An[1,2]*, Yingying Chen[1], Chuanxi Zhao[3,1], Chenhui Li[1], Yongjie Wang[1], Weicai Wang[1], Jiule Li[1], Guangjian Wu[1], Lin Bai[1], Fan Zhang[1], Tandong Yao[1]

[1]State Key Laboratory of Tibetan Plateau Earth System, Environment and Resources (TPESER), Institute of Tibetan Plateau Research, Chinese Academy of Sciences, Beijing 100101, China

[2] School of Science, Tibet University, Lhasa 850011, China

[3] College of Earth and Environmental Sciences, Lanzhou University, Lanzhou 730000, China

*Correspondence to*: anbaosheng@itpcas.ac.cn; yangww@itpcas.ac.cn

**Abstract:** The Tibetan Plateau and its surroundings have recently experienced several catastrophic glacier-related disasters. It is of great scientific and practical significance to establish ground-based early warning systems (EWS) to understand the processes and mechanisms of glacial disasters and warn against potential threats to downstream settlements and infrastructure. However, there are few sophisticated EWSs on the Tibetan Plateau. With the support of the Second Tibetan Plateau Scientific Expedition and Research Program (STPSER), an EWS was developed and implemented in the Sedongpu Valley, southeastern Tibetan Plateau, where repeated river blockages have occurred due to ice/rock collapse debris flow. The EWS collected datasets of optical/thermal videos/photos, geophone waveforms, water levels, and meteorological variables in this sparsely populated zone. It has successfully warned against three ice-rock collapse - debris flow - river blockage chain events, and seven small-scale ice-rock collapse- debris flow events. Meanwhile, it was found that the low-cost geophone can effectively indicate the occurrence and magnitude of ice/rock collapses by local thresholds, and water level observation is an efficient way to warn of river blockages. Our observations showed that there were no immediate meteorological triggers for the ice-rock collapses and associated debris flows. Several factors, such as the volume and location of the collapses and the percentage of ice content involved, influence the velocities of debris flows and the magnitude of river blockages. There are still two possible glaciers in the study area that are at risk of ice collapse. It is worth monitoring their dynamic changes using high-resolution satellite data and the ground-based EWS to safeguard the surrounding hydrological projects and infrastructure in this transboundary region.

## 1. Introduction

The rate of air temperature warming (0.42 °C per decade) in the Tibetan Plateau and surrounding regions is approximately twice the global average (Yao et al., 2022). As a result, glaciers on the Tibetan Plateau have experienced accelerated mass loss (Bhattacharya et al., 2021; Hugonnet et al., 2021), thermal changes (Gilbert et al., 2020), and dynamic changes (Dehecq et al., 2019), contributing to glacier instability that triggered ice collapses and glacier surges (Evans et al., 2021). Glacier-related disasters have been reported in the last decade, such as the Karayaylak glacier surge in the eastern Pamir (Zhang et al., 2022), the collapse of the Aru Co twin glaciers in the western Tibetan Plateau (Kääb et al., 2018; Tian et al., 2017), glacier detachments and ice-rock collapse in the Petra Pervogo range in Tajikistan (Leinss et al., 2021), repeated ice collapses and surges at Mt. Amney Machen in the northeastern Tibetan Plateau (Paul, 2019), massive ice-rock collapses at Chamoli in the Indian Himalayas (Shugar et al., 2021), and the massive glacier detachment and repeated ice-rock collapses in the Sedongpu valley in the southeastern Tibetan Plateau (Li et al., 2022a; Kääb et al., 2021; Zhao et al., 2022).

In high-altitude alpine regimes, ice-rock masses have high potential energy, and the ice-rock collapses typically cause cascading hazards, such as debris flows (Peng et al., 2022), river blockages (Chen et al., 2020), and glacier lake outburst floods (Allen et al., 2022), threatening downstream settlements and infrastructure. For example, the detachment of the Sedongpu Glacier in October 2018 blocked the Yarlung Tsangpo River for ~60 h, damaging or threatening roads and hydropower plants and leading to the relocation of more than 6000 local residents (Chen et al., 2020). The Chamoli rock-ice collapse and the associated debris flow in 2021 resulted in more than 200 deaths and destroyed two hydropower plants downstream (Shugar et al., 2021).

Some precursory signs, such as abnormal changes in surface velocity, surface crevasses and glacier ice relocation, can be captured before a glacier collapse using remote sensing methods (Kääb et al., 2018). However, satellite data is subject to favourable weather conditions and revisit cycles. Given the short duration of glacier collapse (*e.g.* about 5 minutes for the 2021 collapse in Sedongpu valley and 2-3 minutes for the 2016 Aru collapse: Zhao et al., 2022; Kääb et al., 2018), it is difficult to provide timely warnings of glacier collapses and assess their impacts using remote sensing only. Ground-based early warning systems (EWS) provide a real-time monitoring dataset for warning against catastrophic disasters. However, the installation and maintenance of EWS in sparsely populated regions generally faces many challenges such as the instrument transport and logistics in high altitude mountainous areas, the harsh extreme weather conditions, the power supply and data transmission in cloudy and rugged regions, the reliability and compatibility of different sensors, and the sustainable funding. Some ground-based EWSs for glacial lake outburst floods have been implemented on the Tibetan Plateau and at high-elevation (Haemmig et al., 2014; Huggel et al., 2020; Petrakov et al., 2012; Wang et al., 2022). However, few successful warnings have been reported (Massey et al., 2010), and it is difficult to assess the reliability and transferability of such EWSs partly due to the rare reoccurrence of catastrophic disasters (Stähli et al., 2015).

The repeated ice-rock collapses and the induced river blockages in the Sedongpu Valley in the southeastern Tibetan Plateau (An et al., 2022; Chen et al., 2020; Kääb et al., 2021; Li et al., 2022a; Tong et al., 2019; Zhao et al., 2022) motivated us to

establish the ground-based EWS. The EWS was implemented to provide real-time warning signals to the local government and to study the process and mechanism of the catastrophe from the Sedongpu Valley. The aim of this study is to introduce the structure of three EWSs installed near/inside the Sedongpu Valley, to analyse the performance of different monitoring signals (*e.g.* water level, geophone waveform, meteorological variables, optical/thermal images) on warning the occurrence and process of different types of ice-rock collapses (ice-rock mixed or rock-dominated events) and finally to discuss the possible monitoring

priority and challenge for ice-rock disasters on the Tibetan Plateau. Such work on EWS is not only helpful for understanding the process and mechanism of glacier-related disasters but also paves the way for establishing similar EWSs in other high-risk regions of the Tibetan Plateau.

## 2. Study region and historical ice-rock collapses

### 2.1 Study region

The study region is located in the Namcha Barwa-Gyala Peri massif on the southeastern Tibetan Plateau (Fig. 1a). This region is characterized by high tectonic activity, wide topographic variations, a deepened incision by the Yarlung Tsangpo River, influence of the Indian summer monsoon, the concentration of temperate glaciers and deposition of thick Quaternary glacial till. The Namcha Barwa-Gyala Peri massif has experienced the fastest uplift rate and highest denudation rate in the world (Enkelmann et al., 2011; King et al., 2016), and is, therefore, an earthquake-prone area that produced the 1950 Tibet-

Assam earthquake (Mw 8.6). There are two peaks over 7000 m above sea level (m asl): Mt. Namcha Barwa (7782 m asl) and Mt. Gyala Peri (7294 m asl). The Yarlung Tsangpo River carves a deep gorge with a vertical elevation difference of more than 5000m. The Indian summer monsoon intrudes via the Yarlung Tsangpo Canyon, resulting in the longest annual rainy season on the Tibetan Plateau (Yang et al., 2013). Total precipitation in Medog, about 60 km from the Sedonpu Valley, was more than 1200 mm in 2019-2020, with 56.6% falling in June-September and 32.4% in the spring season (March-May) (Li et al.,

2022b). The combination of high relief and abundant monsoonal precipitation results in 141 modern temperate glaciers, with a total glaciated area of 263 km$^2$ around these two peaks (Arendt et al., 2017). The competition between rapid uplift and glacial erosion around the Namcha Barwa-Gyala Peri massif contributed to the formation of thick Quaternary glacial deposits (Hu et al., 2020; Montgomery et al., 2004). These specific tectonic, climatic and topographic conditions have led to massive prehistoric and modern catastrophes and river blockages (Chen et al., 2020; Montgomery et al., 2004; Zhang, 1992).

The Sedongpu basin covers an area of 66.9 km$^2$ and has a length of ~11 km, with elevations ranging from the Mt. Gyala Peri at 7294 m asl to the valley outlet at 2700 m asl (Fig. 1b). According to the Randolph Glacier Inventory (RGI) 6.0 (Arendt et al., 2017), there are five major glaciers in the valley. The largest glacier is the Sedongpu Glacier (RGI60-13.01428) with an

area of 5.0 km², most of which was detached in October 2018 (Kääb et al., 2021). The glacier surface is covered with a thick layer of debris. The bedrock consists mainly of Proterozoic marble and gneiss (Chen et al., 2020).

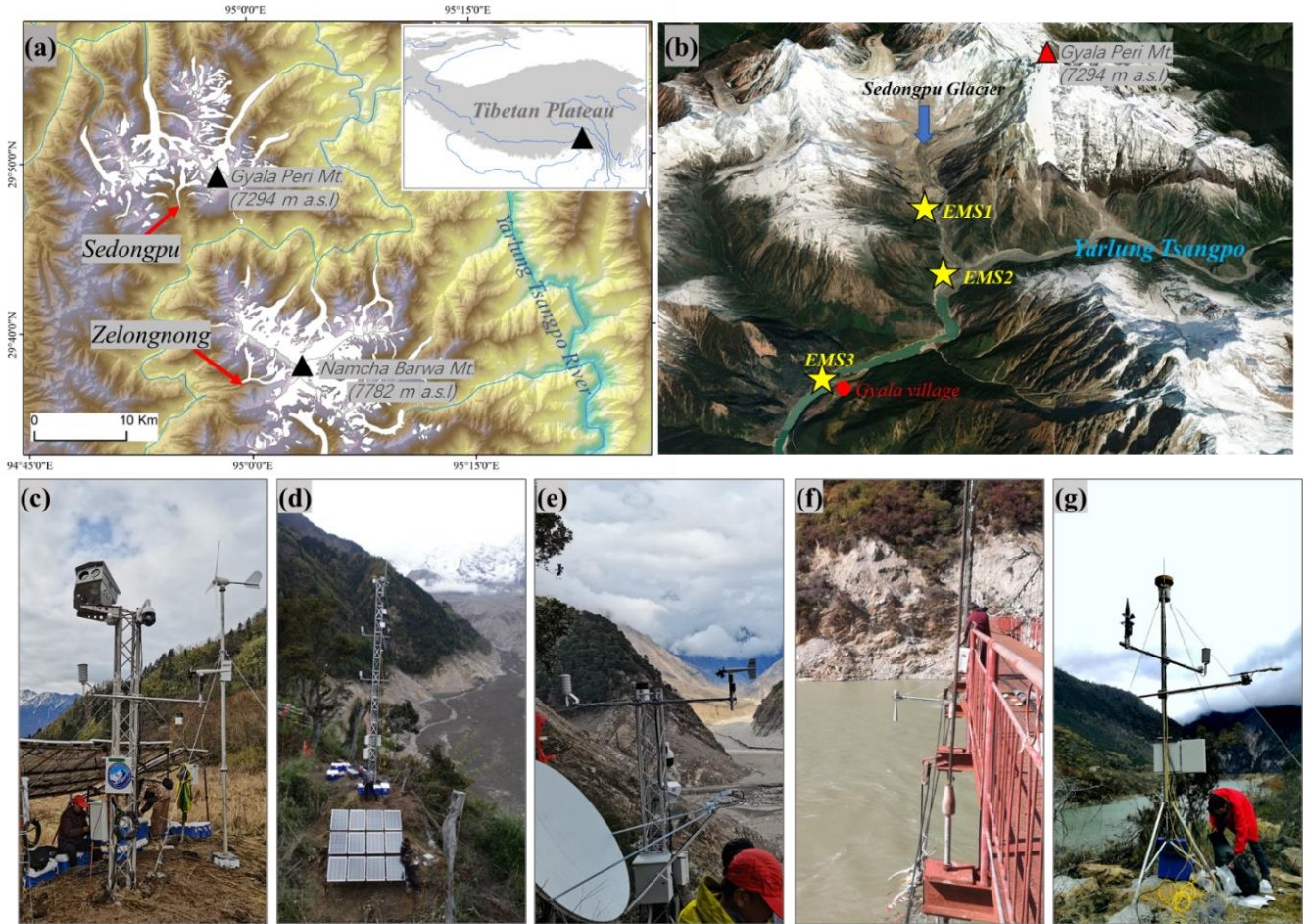


**Figure 1.** Study region and early warning systems. (a) Glacier distribution around the Mt. Namcha Barwa and the Mt. Gyala Peri in the southeastern Tibetan Plateau with the Sedongpu Glacier and Zelongnong Glacier; (b) @Goolge Earth image showing the 3D topography of the Sedongpu Valley and the locations of three early warning systems: EWS1-3 (yellow stars); (c) the EWS1 inside the Sedongpu Valley for ice collapse observations; (d) the EWS2 of the 10 m observation tower, which

was installed near the Sedongpu Valley outlet but was destroyed by the massive ice-rock collapse on 22 March 2022; (e) the rebuilt EWS2 near the Sedongpu Valley outlet to warn the blockage of the Yarlung Tsangpo River; (f, g) the EWS3 of the radar water level sensor and the automatic weather station (AWS) with pressure water level sensor near the Gyala village.

## 2.2 Historical ice-rock collapses around the Mt. Gyala Peri and the Mt. Namcha Barwa

### Historical ice-rock collapses in the Sedongpu Valley near Mt. Gyala Peri

Both the well-developed vegetation inside the Sedongpu Valley and the vegetation-covered dam near the outlet of the Sedongpu Valley during the period from the 1970s to the 2010s indicated that no massive ice collapse-induced debris flows had occurred during the past 40 years (Li et al., 2022a).

Since 2014, ice-rock collapses have occurred frequently in the Sedongpu Valley. Satellite images show that a debris flow destroyed the valley floor forest and vegetation-covered dam between June 2014 and October 2014 (Tong et al., 2019; Li et
al., 2022a). The height difference between the two digital elevation models (DEMs) between 2013 and 2015 showed that the total collapse volume from the northern ridge of Mt. Gyala Peri was ~4 $Mm^3$ (Li et al., 2022a). From October 2017 to October 2018, repeated ice-rock collapse-induced debris flows occurred in October 2017, November 2017, December 2017, January 2018 and July 2018 (Tong et al., 2019). Among these debris flows, the event on 22 October 2017 was the most severe, destroying a large area of vegetation in the valley. Based on the differences in DEMs in 2015 and in 2018, the total volume of
these collapses was more than 50 $Mm^3$ on the northern ridge of Mt. Gyala Peri (Kääb et al., 2021).

On 16 October 2018, a total of 130 $Mm^3$ of low-angle debris-covered glacier detached from the Sedongpu Valley (Kääb et al., 2021; Li et al., 2022a).This event block the Yarlung Tsangpo River for ~60 h and the river level increased about 75m above the original level, which damaged two upstream bridges and inundated dozens kilometres of roads and power supply facilities and forced the evacuation of at least 6000 local resident (Chen et al., 2020). The blocked dam was overtopped on 19
October, with the peak breaching flow as large as 32000 $m^3$/s and damaged the downstream hydropower station. On 22 March 2021, massive ice-rock collapses totalling 50 $Mm^3$ occurred in the Sedongpu Valley, producing a mobile debris flow that temporarily blocked the Yarlung Tsangpo River, leading to the inundation of road to Gyala village (Zhao et al., 2022).

### Historical ice-rock collapses in the Zelongnong valley near the Mt Namcha Barwa

The Zelongnong Glacier (RGI60-13.01428), 20 km south of the Sedongpu Glacier, is located near Mt. Namcha Barwa
and has a total glacier area of 9.5 $km^2$, ranging from 3819 to 6931 m asl (Fig. 1a). Both glacial and fluvial deposits near the outlet of the Zelongnong Valley indicated that this glacier has repeatedly advanced or collapsed since the last glacial maximum (Hu et al., 2020; Montgomery et al., 2004). Four modern glacier collapses have been reported on the Zelongnong Glacier. River blockages have been reported to have occurred in 1950, 1968 and 1984 and the glacier collapse in 1950 engulfed the village of Zhibai and lead to the death of to 97 villagers (Zhang, 1992). In 2020, a total of 1.14 $Mm^3$ of ice-debris mixture
produced a high-speed debris flow and partially blocked the Yarlung Tsangpo River and damaged the Zhibai Bridge (Peng et al., 2022).

## 3. The ground-based EWSs near the Sedongpu Valley

Since 2019, the EWSs have been progressively built along the Sedongpu Valley. The EWSs consist of three parts (EWS 1-3) with different monitoring instruments and scientific functions (Fig. 1c-g and Table 1).

**3.1 EWS1 inside the Sedongpu Valley**

In May 2022, a 4 m observation tower with various monitoring sensors was constructed on the right bank of the glacial terraces at 3308 m asl in the Sedongpu Valley (Fig. 1b and c). Owing to logistical inaccessibility, all instruments were transported by helicopters. The objective of EWS1 is to monitor the location, magnitude and process of the ice-rock collapse inside the Sedongpu Valley. The overall monitoring system consists of the following five parts:

i) Panoramic surveillance of the Sedongpu Valley. A 360° rotatable high-definition dome camera was mounted on the tower and aimed at the Sedongpu Valley to provide daytime video surveillance and high-frequency timing photographs. Because of the limited capability of satellite transmission, the video of the disaster process was stored locally on the hard disk and only retrieved remotely by command. Photographs taken hourly from 8 a.m. to 8 p.m. were regularly transmitted to the server of the National Tibetan Plateau Data Centre (https://data.tpdc.ac.cn/home). Such panoramic monitoring is

used to monitor the process of the ice and rock collapse along the Sedongpu Valley.

ii) Targeted monitoring of the collapsed area. Optical and thermal cameras were used to monitor the dynamic changes in rocks and glaciers on the northern ridge of Mt. Gyala Peri, where ice and rock collapses have occurred repeatedly over the past five years (Kääb et al., 2021; Zhao et al., 2022). The video of the disaster process can be retrieved remotely by command. Hourly photographs were transmitted to the server via satellite. Both optical and infrared photographs are

often affected by the heavy cloud cover and rainfall in this high-altitude region during the monsoon season, making them unsuitable as the real-time warning indicators. This targeted monitoring was designed to identify the location and magnitude of the recurrent collapses by comparing the pre- and post-event photographs.

iii) Surface vibration from repeated collapses. Ice-rock collapses and highly mobile debris flows generally produce significant surface vibrations. A 5 Hz three-component (XYZ vector) SmartSolo geophone (recording ground movements

and converting them into voltage) was used to record the surface vibrations from collapses and debris flows. Owing to a large amount of useless waveform data and the limited capability of satellite transmission, a threshold for data transmission was adopted in the field. During the installation of EWS1, we witnessed several small-scale snow-ice collapses in daytime and several small-scale collapses occurred at midnight were recorded by the geophone. The corresponding amplitude of the three-component waveform was generally greater than three when the collapse occurred.

Therefore, if any XYZ vector was greater than three, the 200-second waveform data before and after the threshold were transmitted automatically to the server.

iv) Meteorological records inside the valley. With regard to possible triggers of extreme weather conditions for ice-rock

collapses and the lack of meteorological data in the Sedongpu Valley, meteorological variables were recorded every 30min using the Campbell datalogger and were transmitted to the server. Air temperature ($T_{air}$) and relative humidity (RH) were measured using the temperature and relative humidity probe (Vaisala HMP155A). Precipitation was measured using a rain gauge (Texas TE525MM). Wind speed and direction were recorded using a wind monitor (RM Young 05106).

v)  Data transmission and power supply. All datasets were transmitted via the Asia-Pacific satellite (Asiasat 7). Owing to the heavy monsoon cloud cover and less efficient solar power generation in this region, power is supplied by both solar panels and wind (Fig. 1c).

## 3.2 EWS2 near the Sedongpu Valley outlet

Ice-rock collapses generally produce debris flows that are deposited near the Sedongpu Valley outlet and then block the Yarlung Tsangpo River (Chen et al., 2020; Tong et al., 2019). The EWS2 was installed near the valley outlet to monitor the river blockage. In fact, a 10 m integrated observation tower, equipped with time-lapse optical and thermal cameras and various meteorological variables (wind speed/direction, air temperature, relative humidity, four components of radiation, rainfall, atmospheric pressure), was installed 150 m above the valley floor at the valley outlet in October 2019 (Fig. 1d). However, this EWS2 was destroyed by the collapse-induced mass flow on 22 March 2021, which had overtopped the 200 m high hill at the basin outlet and stripped the surrounding slope of vegetation (Zhao et al., 2022).

In May 2021, the EWS2 was re-established in the upstream region to avoid possible destruction by debris flows of similar magnitude as in March 2021 (Fig. 1e). Similar to the monitoring equipment at EWS1, EWS2 was equipped with optical and thermal cameras to monitor the status of the Sedongpu Valley outlet, a three-component SmartSolo geophone with a transmission threshold of 10 to indicate the power of debris flows and various meteorological sensors (Table. 1). The video was stored locally on the hard disk, and hourly photos were transmitted to the service via satellite.

## 3.3 EWS3 at the Gyala village

The main objective of EWS3 was to provide real-time early warning information of river blockage through changes in the water level at Gyala village, which is located ~6 km upstream of the Sedongpu Basin. The water level will soon rise when the Yarlung Tsangpo River is partially or completely blocked by debris flow from the Sedongpu Valley. Therefore, the water level is a very important early warning indicator. Two different types of water level sensors were used to provide sufficient redundancy and to avoid possible sensor failures. A Campbell CS477 radar water-level sensor (10-minute interval) and a Campbell CS451 pressure transducer (5-minute interval) were installed separately in October 2019 and May 2021 to provide real-time records of the water level in case of river blockages (Fig. 1f).

Considering the relatively open topography near Gyala village, an automatic weather station was also installed (Fig. 1g) for long-term monitoring of the regional background of climate change by measuring air temperature, relative humidity, wind

speed/direction, precipitation, and four components of radiation including incoming shortwave radiation ($S_{in}$), reflected shortwave radiation ($S_{out}$), incoming longwave radiation ($L_{in}$), outgoing longwave radiation ($L_{out}$) using a CNR4 net radiometer, and air pressure using a Vaisala PTB110 barometer (Table. 1).

## 4. Performance of the EWS

The warning signals were automatically sent to the mobile phones of two responsible experts at STPSER whenever any geophone waveform or water level exceeded the specified thresholds. Subsequently, the experts checked the multi-parameters including the real-time photos, videos, water level and the geophone waveform characters to determine the glacier and river status and informed the local government. Since the establishment of the EWS, the system has successfully detected three ice-rock collapse-debris flow-river blockage chain disasters and alerted the local government of early disaster management. In addition, the system monitored seven small-scale ice-rock collapse-debris flow events that did not result in river blockage, but the debris mass flow approached the Yarlung Tsangpo River. Here, the performance of the EWS is briefly described and the effectiveness of early warning indicators is discussed, such as the geophone waveform and water level.

### 4.1 Massive ice-rock collapse and river blockage on 22 March 2021

On 22 March 2021, a massive ice-rock collapse occurred from the northern ridge of Mt. Gyala Peri, and the resulting mass flow temporarily dammed the Yarlung Tsangpo River. Based on a comparison of pre- and post-event DEMs, the total collapse volume was estimated to be ~50 $Mm^3$ (Zhao et al., 2022). The water level sensor at EWS3 sent an automatic warning message due to a sudden 2.2 m rise in the river level between 23:50 on 22 March and 00:00 on 23 March (Fig. 2). The water level continued to rise at a rate of 0.6 - 0.8 m/hour, rising by a total of 11 m before stabilising at approximately 18:00 on 23 March 2021. This collapse turned into a highly mobile mass flow which destroyed the EWS2 installed near the Sedongpu Valley outlet (Fig. 1d). Unfortunately, the video of this chain disaster was not recorded.

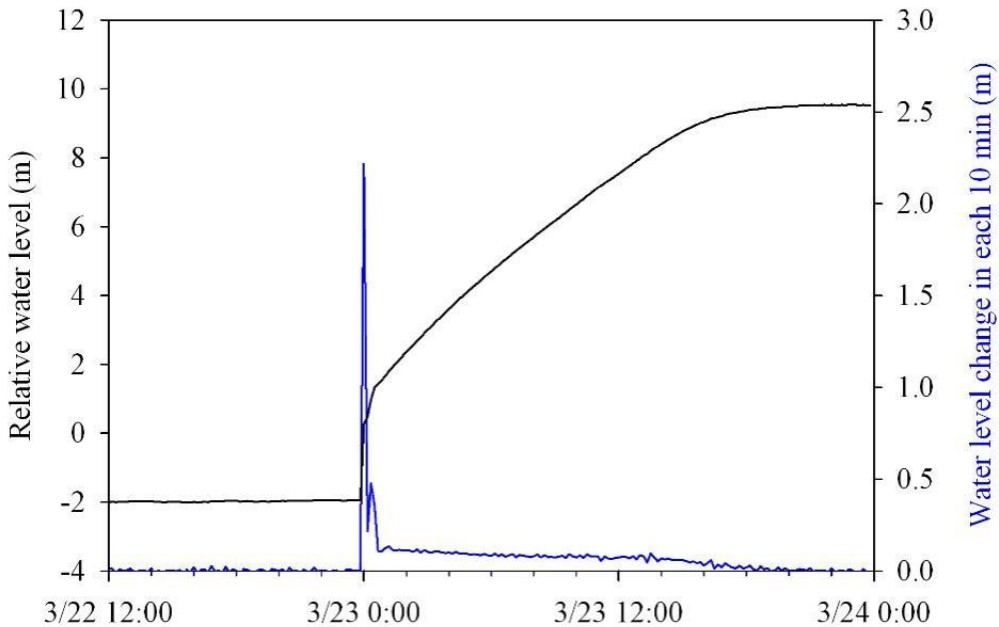

**Figure 2.** A total water level rise of 11 m (black line) caused by the damming of the Yarlung Tsangpo River on 22 March
2021, with the rate of water level rise every 10 minutes (blue line)

### 4.2 Ice-rock collapses and river blockage on 14 May 2022

On 14 May 2022, the geophone at EWS1 warned that repeated collapses occurred in the Sedongpu Valley (Fig. 3a). The
most pronounced shaking occurred at approximately 10:00 and 12:00, with both waveform amplitudes exceeding 20. However,
both EWS2 near the Sedongpu Valley outlet and EWS3 near Gyala village did not show river blockage and water level rising
before midnight on 14 May 2022. Between 23:50 on 14 May and 00:00 on 15 May, the water level sensor at EWS3 sent a
warning message, and the water level rose by a total of 4 m before stabilising at around 20:00 on 15 May 2022 (Fig. 3b).
Infrared imagery at EWS2 showed that the Yarlung Tsangpo River was temporarily completely dammed and then released,
consistent with the rapid rise and fall in the water level (Fig. 4ab). The AWS at EWS1 recorded a total rainfall of 31.6 mm
during the period from 14 May to 17 May with intensive rainfall (7 mm) occurring between 01:00 and 04:00 on 15 May (Fig.
3b). Overall, the ice-rock collapse and intensive rainfall provided favourable conditions for the formation of a diluted debris
flow, resulting in the temporary blockage of the Yarlung Tsangpo River (Fig. 4a, b).

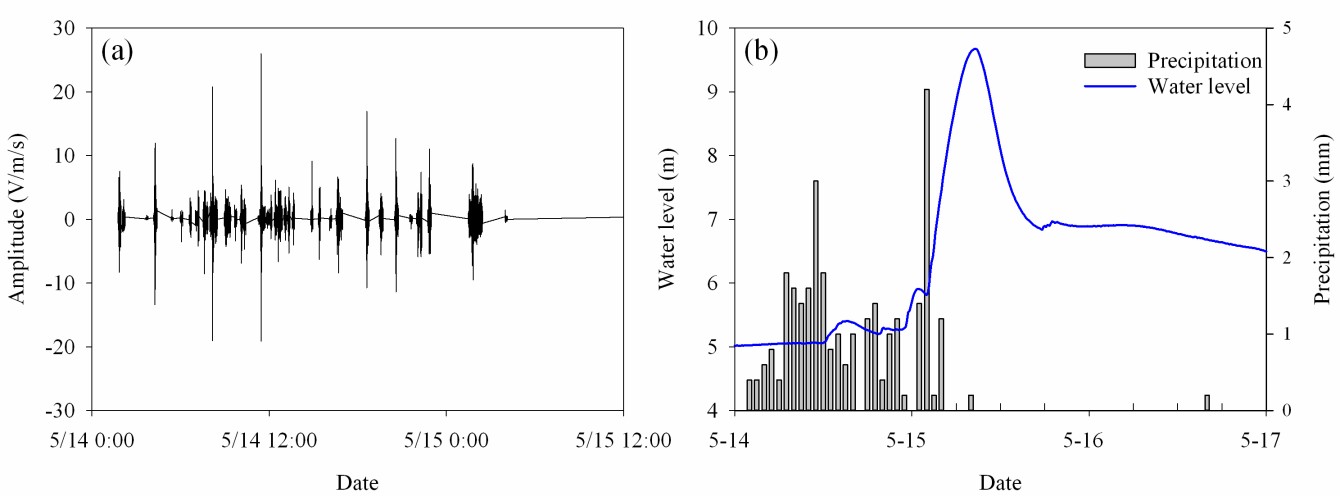

**Figure 3.** The recorded geophone waveform at EWS1 due to frequent ice-rock collapses on 14 May 2022 (a), and the rising water level of the Yarlung Tsangpo River due to the blockage on 15 May 2022 along with the hourly precipitation (b)

225

A comparison of both optical and infrared photographs before and after the ice-rock collapse showed that repeated collapses occurred on the northern ridge of Mt Gyala Peri (Fig. 4c-f), which is close to the locations of previous massive ice-rock collapses in 2017 (Kääb et al., 2021) and 2021 (Zhao et al., 2022). In fact, the ridge was heavily glaciated, so these repeated events were mixed ice-rock collapses. The delayed formation of debris flows after ice-rock collapses (about 20 hours

230 later than the first collapse) suggests that it takes some time for the collapsed ice to melt, which agrees well with the delay between the observed ice collapse in October 2019 and its delayed debris flow (Zhao et al., 2022).

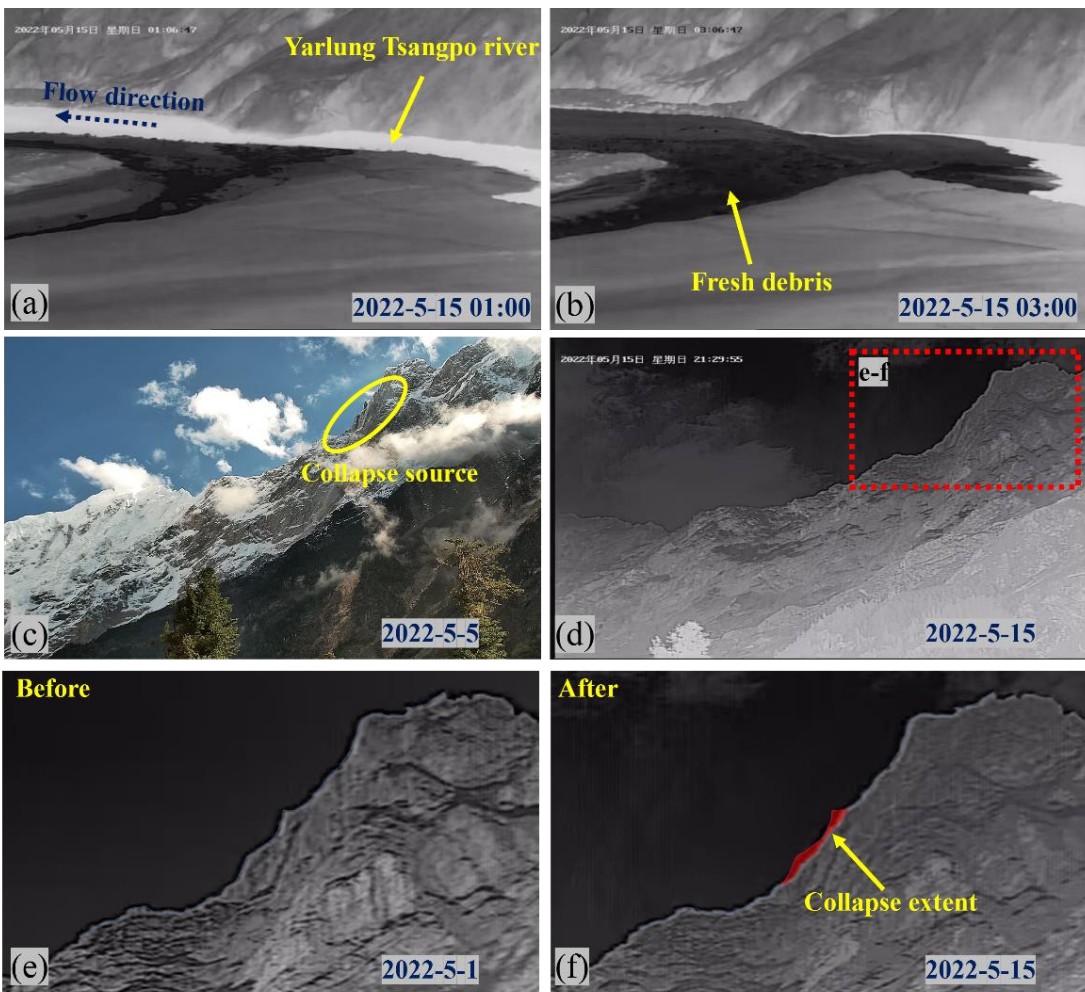

**Figure 4.** The debris-induced blockage of the Yarlung Tsangpo River (a, b) and the optical and thermal images showing the topographic conditions at the targeted monitoring of the collapsed area (c, d) with the zoomed region (red dashed rectangle) shown in e-f, and the change of ridgeline before and after the repeated ice-rock collapses on 14 May 2022 and the extent of the collapse source are outlined in transparent red.

### 4.3 Repeated ice-rock collapses on 11 August 2022

On 11 August 2022, the geophone at EWS1 warned of repeated collapses in the Sedongpu Valley (Fig. 5a). The first collapse occurred at 16:26, with a waveform amplitude greater than 20, and lasted approximately 5-7 minutes. The following six repeated collapses occurred within the next eight hours. Among these seven repeated collapses, the waveform showed that the greatest energy was released by the first collapse, with a maximum amplitude greater than 25, and the longest collapse process lasted approximately 84 mins during the third event (Fig. 5a).

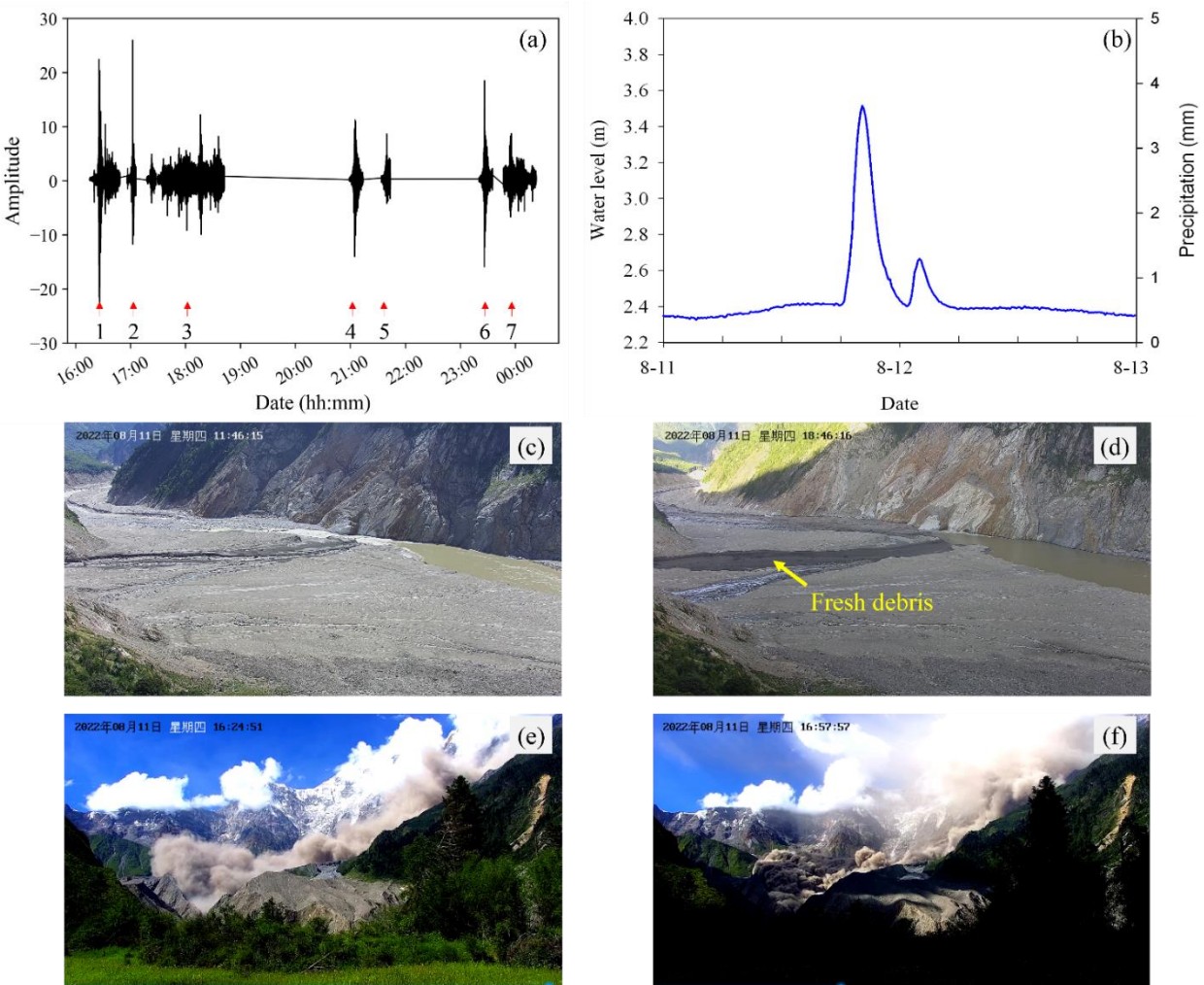

**Figure 5. The recorded waveform of seven collapses on 11 August 2022 (a) and the water level rise by small-scale debris flow and precipitation condition (b), and the photos taken before and after the occurrence of fresh debris flow into the Yarlung Tsangpo River (c,d), and video screenshots of two rock collapses at 16:24 and 16:57 on 11 August 2022.**

The warning system at EWS3 indicated that the water level began to rise at around 18:30 and rose by 1.1 m at around 20:00, and then started to fall back rapidly (Fig. 5b). Photographs of EWS2 also show that small-scale fresh debris flowed into the Yarlung Tsangpo River, causing partial river blockage (Fig. 5c and 5d). Comparison between the post- and pre-event optical and infrared imagery at EWS1 showed that the collapsed sources were different from previous ice-rock collapses (Fig. S1 in the Supplement). Two collapsed regions are mainly concentrated a few hundred meters below the mountain ridge, with

a few glacier distributions. This evidence suggests that lower ice content within collapsed materials prevent the formation of highly mobile debris flows and thus limit the magnitude of river blockage.

Owing to the sunny weather conditions, the optical video system at EWS1 successfully captured the entire process of the two collapses that occurred during the daytime (Fig. 5e and 5f, Supplementary videos). Both videos of the two collapses showed that the highly mobile collapsed material resulted in dark yellow dust in the Sedongpu Valley. The colours of both collapses differed from the observed ice collapse recorded in October 2019 (Zhao et al., 2022). This indicates that the majority of the collapsed material on 11 August 2022 was composed of rock. Second, the combination of the video and geophone waveform showed that the size of the collapses was impressive when the amplitude of the geophone waveform at EWS1 was greater than 20. In addition, the onset and duration of the collapse recorded by the geophone were in agreement with the video recordings. These matches demonstrate that the geophone waveform at EWS1 can be used to reflect the onset and magnitude of repeated collapses in the Sedongpu Valley.

## 4.4 Warning indicators and their applications

As shown by the above-mentioned ice/rock collapse-debris flow-river blockage events, water level monitoring proved to be an effective means of warning against river blockage. The water level monitoring intervals were 5 mins and 10 mins at EWS3. Based on the analysis of the water level data and three previous blockage events, water levels rising at three-level thresholds of 20, 25, and 30 cm per 10 mins were selected. Meanwhile, the real-time camera and hourly photos at EWS2 were also used as complementary data to confirm river blockages.

Previous retrospective studies have shown that seismic observations are helpful in reconstructing the processes of both glacial lake outburst floods (Cook et al., 2018; Maurer et al., 2020) and ice-rock collapses (Bai et al., 2023; Cook et al., 2022; Tiwari et al., 2022). The above-mentioned ice-rock collapses in the Sedongpu Valley further demonstrated that the *in-situ* geophone waveform can be effectively used to warn against the occurrence and magnitude of ice-rock collapses. When the X-axe amplitude of the geophone waveform at EWS1 was approximately 20, the magnitude of the ice-rock collapse was similar to that of the 11 August 2022 event (Fig. 5e and f). The geophone records at EWS1 show that there are a total of 12 events with waveform amplitudes greater than 20 from May 2022 to December 2022 (Fig. 6). Combined with the infrared and optical photographs at the Sedongpu Valley outlet, nine of the twelve events were confirmed to correspond to the occurrence of debris flows that arrived at the Sedongpu Valley outlet but did not significantly block the river (Figs. S2-8 in the Supplement). Only three events, including the Mw 5.6 earthquake centred at Medog on 10 November 2022, did not correspond to the debris flow near the valley outlet. Therefore, when the geophone amplitude exceeds 20 in EWS1, there is a high probability that collapse-induced debris flow will occur in the Sedongpu Valley. The geophone amplitude of 20 was selected as the first-level warning threshold. Due to no catastrophic events occurred in 2022, the EWS defined temporally two times (40) and three times (60) of the first-level threshold as the second and third-level warning thresholds, which would be optimized by further observations.

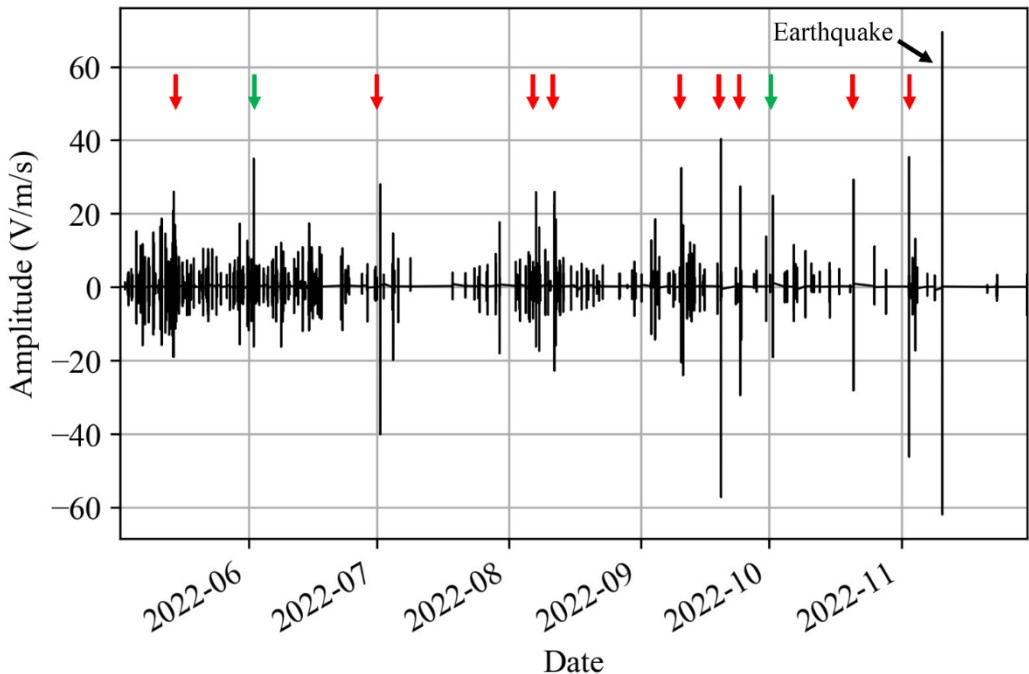

**Figure 6.** Geophone waveform inside the Sedongpu Valley during the period from May to December with 12 events of waveform amplitude greater than 20. The red arrow indicates the confirmed collapses and induced debris flows, the black arrow indicates the Mw 5.6 earthquake on 10 November 2022, and the green arrow indicates two abnormal waveforms.

290

The study region is located in an active crustal zone that experiences frequent earthquakes (Li et al., 2022a). A mixed waveform of collapses and earthquakes may result in false warning signals. In fact, the waveforms of ice collapses and earthquakes differ because of different physical processes. Figure 7 shows a comparison of the nine waveforms triggered by the confirmed rock collapses and the Mw 5.6 earthquake on 10 November 2022. The collapse typically lasted for several

295 minutes and produced a continuous waveform (Fig.7 a-i). In contrast, the waveform from an earthquake is usually short-lived owing to the rapid release of rupture energy, *e.g.*, approximately one minute for a Mw 5.6 earthquake (Fig. 7j). In addition, for the two false warnings (maximum amplitude greater than 20) that occurred on 2 June and 1 October (Fig. 6), the detailed waveform analysis can easily exclude such false warnings by assessing their waveform duration and characteristics (Fig. S9 in the Supplement). These two waveforms are abnormally short and different from the typical collapse-induced waveform,

300 indicating the possible noise. Therefore, in addition to the threshold of the maximum amplitude, the duration and characteristics of the waveform would be re-examined by the experts at STPSER and thus adopted as additional information for the ice-rock collapse warning in the Sedongpu Valley.

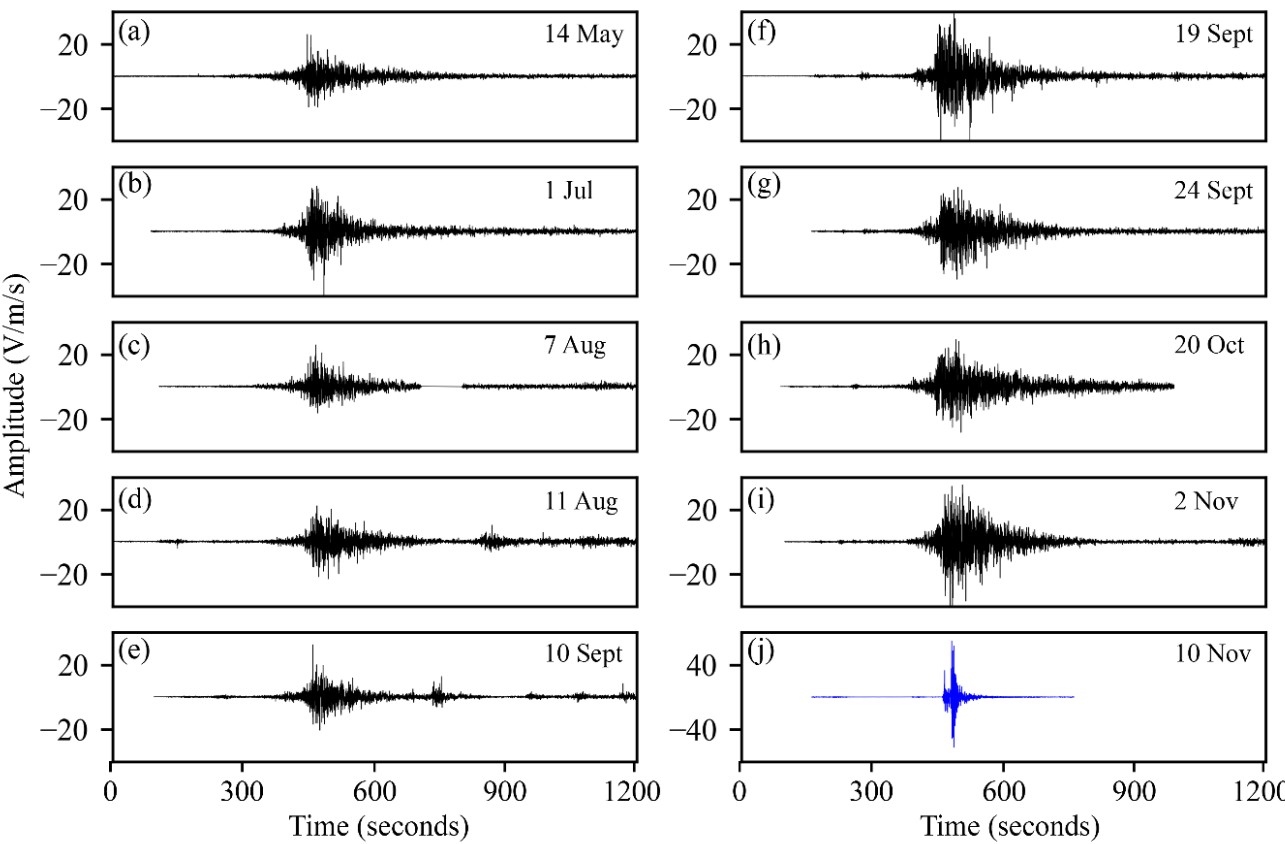

Figure 7. Comparison of the waveforms generated by ice-rock collapses (a-i) and by the earthquake on 10 November 2022 (j). Note the different y-axis limits.

## 5.   Discussions

### 5.1 The relationship between ice-rock collapses and meteorological factors

The EWS showed that at least nine ice-rock collapse-debris flows were found during the period from April to December 2022 in the Sedongpu Valley. Previous studies show that some catastrophic rockslide-debris flows were triggered by heavy rainfall (e.g. Anderson and Sitar, 1995; Yin et al., 2016). The in-situ AWS measurements in/near the Sedongpu Valley provided an opportunity to investigate the relationship between the occurrence of ice-rock collapses and the possible meteorological triggers. Figure 8 shows the variation of meteorological factors such as air temperature, rainfall, wind speed and incoming shortwave radiation and their relevant values in the corresponding ice-rock collapse events (red stars). It is clear that the ice-rock collapses at the Sedongpu Valley could occur under different meteorological combinations (Table S1 in the Supplement).

For example, the weather condition of the ice-rock collapse on 14 May 2022 was characterized by relatively colder weather (8.3 °C), heavy rainfall (22.6 mm), weaker wind speed (0.3 m/s) and lower solar radiation (83.9 W/m²). In contrast, the weather conditions of the ice-rock collapse on 11 August 2022 were warmer (15.4 °C), less precipitation (0.4 mm), moderate wind

320 speed (0.7 m/s) and sunny weather (321.1 W/m²). In addition, it is clear that no weather anomalies (*e.g.* extremely high air temperature or heavy rainfall) prior to the ice-rock events. This suggests that there were no immediate meteorological triggers for these small-scale ice-rock collapses and the associated debris flows in 2022 at the Sedongpu Valley. However, it should be noted that climate change can trigger complex delayed feedbacks in high-mountain regions. Previous studies have shown that the massive ice-rock collapses and glacier detachments occurred concurrently with, or shortly after, positive seasonal air

325 temperature anomalies (Zhao et al., 2022; Bondesan and Francese, 2023). The possible triggers of ice-rock collapse therefore merit further investigation from a variety of perspectives, including the dynamic conditions of temperate glaciers, pre-event seismic activity, and the cumulative role of climate change.

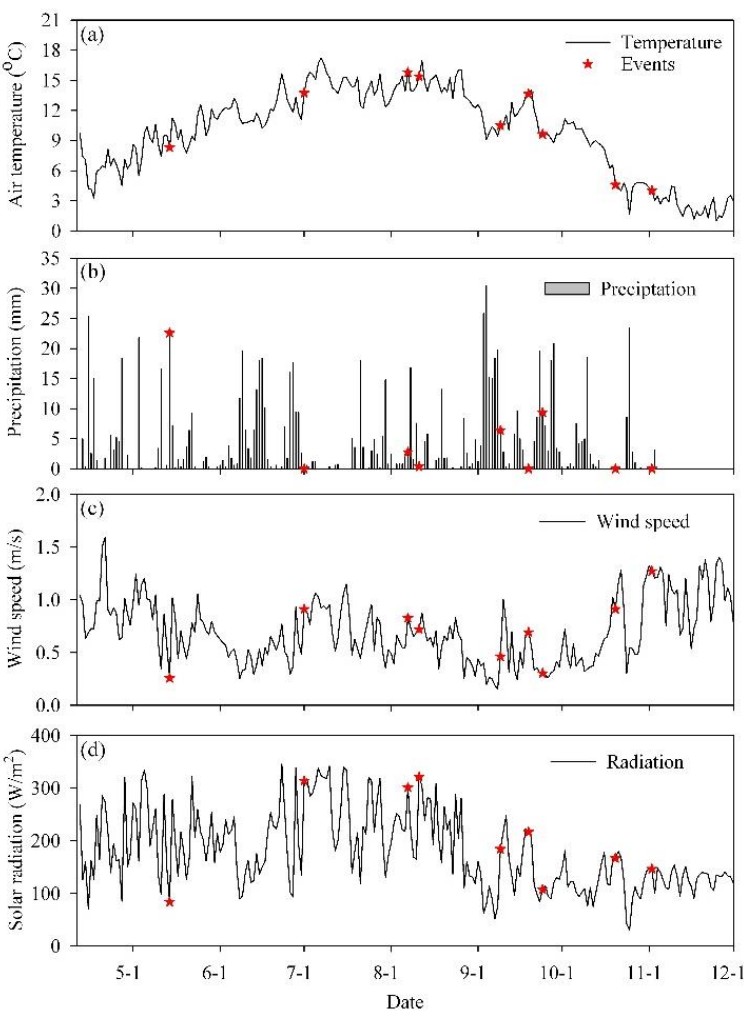

**Figure 8.** The occurrence of repeated ice-rock collapses (red stars) and the time-series of meteorological factors including air temperature (a), rainfall (b), wind speed (c) from EWS1 and the incoming shortwave radiation from EWS3 (d).

## 5.2 Glaciers with potential risks near the Gyala Peri-Namcha Barwa Massif

Repeated ice-rock collapses in the Sedongpu Valley near Mt. Gyala Peri (An et al., 2022; Chen et al., 2020; Kääb et al., 2021; Li et al., 2022a; Tong et al., 2019; Zhao et al., 2022) and the Zelongnong valley near Mt. Namcha Barwa (Peng et al., 2022; Hu et al., 2020; Montgomery et al., 2004; Zhang, 1992) showed high disaster risks in this high mountain region. Based on previous glacier-related disaster studies, some precursors of abnormal glacier dynamics, such as accelerated surface velocity, widening surface crevasse and glacier thickening at the glacier terminus, could be captured prior to glacier collapse using repeated high-resolution remote sensing data (An et al., 2022; Jacquemart et al., 2020; Kääb et al., 2018; Shugar et al., 2021), thus providing first-order discrimination of glaciers at risk.

Based on the published datasets, including glacier elevation change during the period 2010-2020 (Hugonnet et al., 2021) and glacier surface velocity during the period 2017–2018 (Millan et al., 2022), it was found that apart from the detached Sedongpu Glacier, there are still two glaciers (Zelongnong Glacier and the glacier numbered RGI60-13.01430) with possible collapse risks in the study region (Fig. 9). As shown in Figure 9a, the ablation zone of the Sedongpu Glacier has experienced an anomalous glacier thickening (~+2.0 m/yr) over the past decade, which facilitated massive low-angle glacier detachment in October 2018 (Kääb et al., 2021). A similar anomalous ice thickening near the ablation zone (the red rectangles in Figure 8a) was also found at the Zelongnong Glacier (~+0.35 m/yr) and RGI60-13.01430 Glacier (~+0.37 m/yr). In addition, the spatial distribution of the surface velocity also indicated a high glacier displacement in the ablation zone with possible sliding risks on these glaciers (Fig. 9b).

It is therefore worth paying special attention to the dynamic changes of these glaciers by using high-resolution satellite data and ground-based EWS. Enhanced monitoring, particular the surface movement monitoring by using Global Navigation Satellite System (GNSS), is critical for the accessible Zelongnong Glacier, which is very close to the town of Pai with a population of more than 3000, and for the proposed mega hydropower project nearby (Fig.9b). In the event of massive ice-rock collapses or glacier detachment, the potential hazards will be more severe than those from the Sedongpu Glacier and RGI60-13.01430 Glacier because of the limited elevation difference (~70 m) and linear distance (~11 km) between Pai town (2920 m asl) and the Zelongnong Valley outlet (2850 m asl). Therefore, similar to the EWS in the Sedongpu Valley, it is necessary to conduct continuous ground-based monitoring on/near the Zelongnong Glacier to provide early warning for the protection of the surrounding and downstream hydrological projects and infrastructure in this transboundary region.

In addition, special attention should be also paid to the tectonic-climatic interactions near the Gyala Peri-Namcha Barwa Massif. The Namcha Barwa-Gyala Peri has experienced the highest uplift and denudation rates worldwide (Enkelmann et al., 2011; King et al., 2016). The competition between rapid uplift and glacial erosion around the Namcha Barwa-Gyala Peri massif

contributed to the formation of thick Quaternary glacial deposits (Hu et al., 2020; Zhu et al., 2012). With extreme climate events, temperate glaciers in this region become unstable, leading to repeated ice-rock collapses. The combination of excess meltwater from glacier collapse and the thick, unconsolidated glacial deposits is therefore likely to contribute to anomalous glaciofluvial erosion (Kääb et al., 2023). Most of the debris flows and eroded glacial sediments from the high elevations were
eventually transported down by the Yarlung Tsangpo River. Thus, the instability of high-elevation glaciers near the Gyala Peri-Namcha Barwa massif could potentially affect the safety of downstream hydropower dams as well as the ecological systems by changing sediment loads along the transboundary river (Li et al., 2022c).

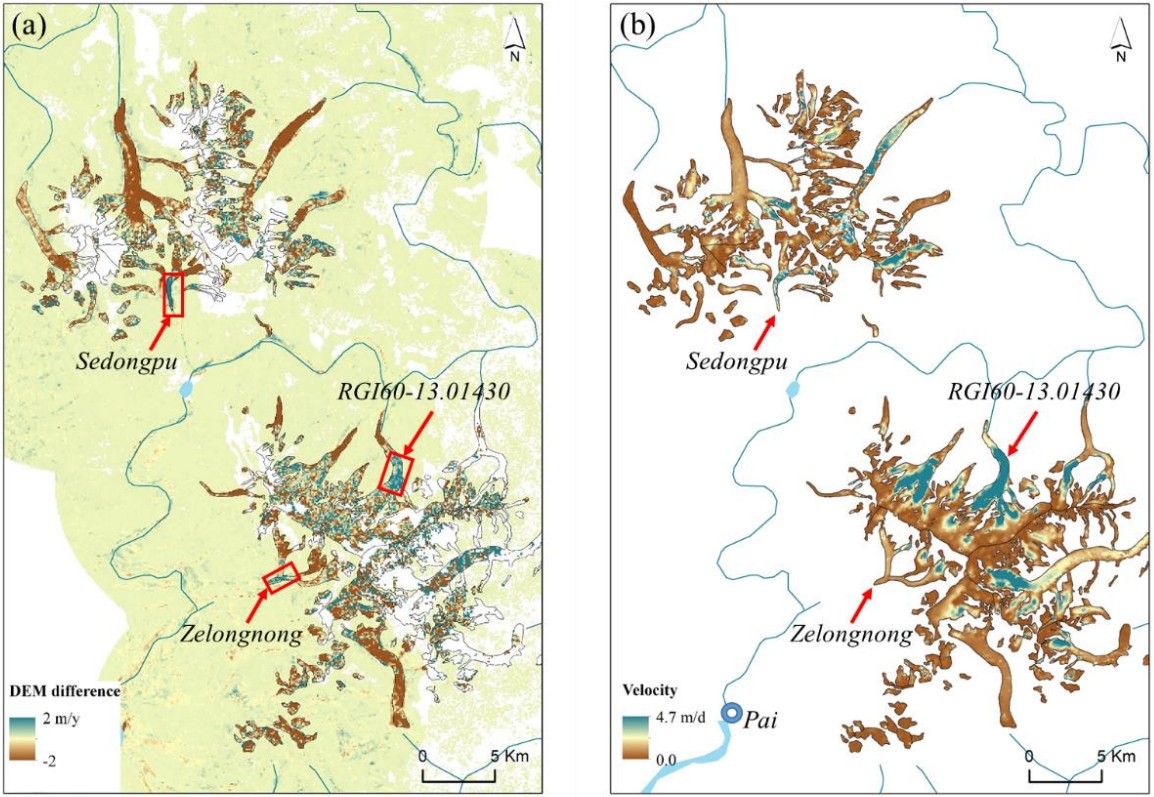

**Figure 9.** Spatial distribution of mean surface elevation change (m/y) during 2010-2020 (Hugonnet et al., 2021) and annual
surface velocity (m/day) in 2017 (Millan et al., 2022), and the locations of the three glaciers with abnormal surface thickening (the red rectangles) and the tower of Pai.

**5.3 The monitoring priority and challenge for ice-rock disasters**

The EWS in this study provides valuable datasets, including optical/thermal video/photographic, seismic, meteorological, and water level data, for monitoring the occurrence of ice-rock collapses and river blockages near the Sedongpu Valley.

However, both *in-situ* and satellite observations have revealed that the collapsed source is generally concentrated at extremely high elevations above 6000 m asl (Kääb et al., 2021; Zhao et al., 2022). Abundant monsoonal moisture and cloud formation at high elevations have prevented the availability of real-time, cloud-free video/imagery to determine the location and magnitude of the ice-rock collapses. As shown in Figures 4c-f, the selected cloud-free photos were taken several days before and after the collapse, delaying the real-time assessment of the collapses. Therefore, new weather-independent monitoring instruments (*e.g.*

all-weather avalanche radar) should be considered for real-time monitoring and for accessing the location and volume of ice-rock collapses in the next updated ground-based EWS at some key locations related to the large infrastructure (*e.g.* mega hydropower stations). Although satellite data sometimes suffer from weather conditions and revisit intervals, repeated comparisons of high-resolution remote sensing data and weather-independent SAR data are also recommended to provide preliminary information on the deformation and instability of rock and ice in high-altitude glaciated regions.

385        Previous studies have shown that seismic observations are an important way to understand the dynamic process of different catastrophic disasters such as ice-rock collapses (Bai et al., 2023; Cook et al., 2022; Tiwari et al., 2022), glacier lake outburst floods (Cook et al., 2018; Maurer et al., 2020), rock landslides (Le Breton et al., 2021), and have important potential for detecting the precursory signals before the event as an early warning by the distant seismic stations up to 100 km from the disaster (Cook et al., 2022). Seismic geophones are widely used in EWS around the world and are one of the main early

warning tools (Massey et al., 2010). Although catastrophic events can be detected in seismic data, the general limits and controls on detection and location remain challenged. Seismic energy generated in surficial catastrophic activities is easily saturated or attenuated with distance (West et al., 2010). For example, the geophone at EWS1 inside the Sedongpu Valley provided a good warning due to its proximity to the source of the collapses (Fig. 1). In contrast, the geophone at EWS2 near the Yarlung Tsangpo River was significantly influenced by river-generated noise and was less sensitive to the small-scale

collapses in 2022 due to its distance from the source regimes. The installation of seismic stations for event detection is therefore a function of the desired minimum detectable event size. The deployment of the optimized seismic stations is crucial to obtain the high-quality signal with less noise ratio in the rugged mountainous regions (Cook et al., 2022). And the reliability of seismic warning should be further confirmed and validated by using multi-parameter observations (e.g. remote sensing data, river gauges, photographs and videos). The concurrent multi-parameter observations (water level, thermal/optical images, and

video) provided the opportunity to validate the occurrence of the above-mentioned nine repeated ice-rock collapses in 2022 and the reliability of the seismic threshold for early warning in the Sedongpu valley. The possible false warns can also be excluded by analysing the duration and characteristics of the waveforms and the corresponding in-situ observations. However, it should be noted that this local threshold (amplitude greater than 20) is empirically appropriate for EWS1. In fact, each phase of the disaster chain has a different seismic signature and a different detection limit at far-field stations (Massey et al., 2010).

The local threshold for seismic warning should be determined by historical records of ice-rock collapse events and validated by the field multi-parameter observations in other hazard regions. In addition, the detection of precursory signals prior to the ice-rock collapse for early warning is very promising but remains a major challenge (Cook et al., 2022; Tiwar et al., 2022).

Automated processing by using innovative technologies such as deep learning methods would be helpful for the effective future warning of catastrophic ice-rock collapse (Tiwar et al., 2022).

410  Moreover, the installation and maintenance of the EWS generally faces the institutional challenges and is necessary to gain the support of the local government and the acceptance of the local population, especially those with strong religious beliefs (Huggel et al., 2020; Sufri et al., 2020). The repeated destruction caused by river blockages improved the local population's and government's knowledge of the ice-rock collapses and the importance of an early warning system in the Sedongpu Valley. As a result, we gained strong support from the local government. The official Disaster Emergency

Management Department is responsible to lunch the alarm initiating evacuation when the serious river blockages are confirmed. The implementation of similar EWS in other high-risk regions should not only focus on the monitoring techniques, but should also pay close attention to the cognitive and response capabilities of the government and local residents.

## 6. Conclusions

  An EWS has been established to monitor ice-rock collapses and river blockages in the Sedongpu Valley in the

southeastern Tibetan Plateau. It consists of three parts with different monitoring sensors and scientific functions: EWS1 for ice-rock collapse warning and EWS2 and EWS3 for river blockage warning. The EWSs provide valuable information on optical/thermal video/photo, seismic, meteorological and water level data transmitted mainly via satellites in this sparsely populated region. The systems successfully detected three ice-rock collapse-debris flow-river blockage chain events and at least seven ice-rock collapse -debris flow events, providing alarming information to the local government.

425  Based on these monitoring efforts, we found that the geophone waveform plays a critical role in warning of collapses in the Sedongpu Valley. The duration and characteristics of the waveform is significantly different from those of an earthquake due to different physical processes. When the geophone waveform amplitude is greater than 20 at EWS1 and the duration of waveform last several minutes, there is a high probability that a collapse-induced debris flow will be generated from the Sedongpu Valley, and the magnitude of the collapses is impressive as shown by the events videotaped on 11 August 2022. In

addition, the collapse-induced river blockage at the Sedongpu Valley outlet can be warned in time by a water level monitoring system. Based on the analysis of these observed disaster events, it was found that several key factors, including the volume/location of the collapses, percentage of ice content in the collapsed material and meteorological conditions, could contribute to the different velocities of the debris flows and different magnitudes of the river blockages. This work on EWS paves the way for the establishment of similar EWSs in other potential collapse regions on the Tibetan Plateau for early

detection of hazards and for effectively reducing risks.

  In addition, based on the previous studies on the precursors of glacier collapse and the published datasets, it was found that, apart from the detached Sedongpu Glacier, there are still two glaciers (Zelongnong Glacier and the RGI60-13.01430 Glacier) with possible collapse risks in the study region. It is worth paying special attention to these dynamic changes using high-resolution satellite data and ground-based EWSs.

**Code/Data availability.** Data in this study are available upon request from the corresponding authors.

**Author contributions.** W.Y., B.A, T.Y, Y.C was responsible for the framework design of the EWS. Z.W., W.Y. and B.A. installed the EWS. W.Y. analysed the data and write the manuscript. G.W., L.B., F.Z., C.Z., C.L, W.W., and J.L. assisted in collecting all data and discussion. Funding acquisition, T.Y. All authors have read and agreed to the published version of the manuscript.

**Declaration of competing interest.** The authors declare that they have no conflict of interests.

**Disclaimer.** Publisher's note: Copernicus Publications remains neutral with regard to jurisdictional claims in published maps and institutional affiliations.

**Acknowledgments**

The study was supported by the Second Tibetan Plateau Scientific Expedition and Research Program (Grant No. 2019QZKK0201), Science and technology projects in Tibet Autonomous Region (XZ202301ZY0028G, XZ202301ZY0022G, XZ202101ZD0014G) and the National Natural Science Foundation of China (Grant Nos. 42271138, 42271312) and Lhasa Earth System Multi-Dimension Observatory Network (LEMON). We would like to sincerely acknowledge and express our deep appreciation to the reviewers for their thorough review of this work.

**Table 1.** Detailed information on the ground-based early warning system installed near the Sedongpu Valley in the southeastern Tibetan Plateau

| Locations | Lat/lon | Elevation (m) | Instruments | Monitoring element and purpose | Frequency | Manufacture | Resolution | Transfer |
|---|---|---|---|---|---|---|---|---|
| Inside the valley | 29.77° 94.92° | 3308 | Optical camera1 | real-time video、daytime photo for the whole valley | Hourly photo from 08:00-20:00 | Hikvision DS | 4 Megapixel | Asiasat 7 |
| | | | Optical camera2 | real-time video、daytime photo for the collapse region | Hourly photo from 08:00-20:00 | IR8000 | 640×512 pixel | |
| | | | Thermal camera | real-time video、night-time photo for the collapse region | Hourly photo from 20:00-08:00 | | 1945×1225 pixel | |
| | | | Geophone | X,Y,Z three-component waveform | 5 HZ | SmartSolo DT-SOLO | Distortion : < 0.15% @12 Hz | |
| | | | AWS | $T_{air}$, RH, rainfall, wind | 10 min | Compbell | - | |
| Valley outlet | 29.75° 94.93° | 2930 | Optical camera | real-time video、daytime photo for river blockage | Hourly photo from 08:00-20:00 | Hikvision DS | 4 Megapixel | Asiasat 7 satellites |
| | | | Thermal camera | real-time video、night-time photo for river blockage | Hourly photo from 20:00-08:00 | STZ100X5 | 640×512 pixel | |
| | | | Geophone | X,Y,Z three-component waveform | 5 HZ | SmartSolo DT-SOLO | Distortion : < 0.15% @12 Hz | |
| | | | AWS | $T_{air}$, RH, rainfall, wind | 10 min | Compbell | - | |
| Gyala village | 29.71° 94.90° | 2780 | Radar water level | water level of Yarlung Tsangpo River | 10 min | Campbell CS477 | 1 mm | 2G cellular network |
| | | | Pressure water level | water level of Yarlung Tsangpo River | 5 min | Campbell CS451 | 0.0035% FS | Beidou satellites |
| | | | AWS | $T_{air}$, RH, $S_{in}$, $S_{out}$, $L_{in}$, $L_{out}$, rainfall, wind, air pressure | 10 min | Compbell | - | |

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
