# Peer review of "Early warning system for ice collapses and river blockages in the Sedongpu Valley, southeastern Tibetan Plateau"

_Natural Hazards and Earth System Sciences, 2023_

## Author Response (AR1)

**Response to Reviewer 1**

Yang et al. Introduced the structure and performance of an Early Warning System installed for monitoring ice collapses and river blockages in the southeastern Tibetan Plateau. The EWS has successfully warn three collapse-river blockage events and sdeven small-scale collapses. They also found that the volume and location of the collapse and the percentage of ice content contribute the different velocity of debris flow and magnitude of rive blockages. Such work represents an important contribution to the cryosphere disasters monitoring and warning on the Tibetan Plateau. Although the manuscript provided good descriptions of EWS and their performance, there are a number of issues below that need to be addressed prior to publication.

Reply: We would like to thank Dr. Zhuang Yu for reviewing and editing this study. We greatly appreciate your insightful comments and respond to each of them below. Responses to the comments of the reviewer are written in blue.

**General Comments:**

The introduction needs more background on the ground-based early warning systems that have already been installed around the world. The author states that few successful warnings have been reported. Could the authors provide some description about the question of EWS performance? In line 51, the author states that there are no EWS for ice collapse on the Tibetan Plateau. Are specific instruments and structure of EWS needed for ice collapse? More explanation should be added to the manuscript.

Reply: Thank you for these insightful comments. The reported EWSs (e.g. Petrakov et al., 2012; Haemmig et al., 2014; Huggel et al., 2020; Wang et al., 2022) were designed for warning the GOLF. However, the occurrence of GOLF is rare. That is the main reason why few successful warnings have been reported (Massey et al., 2010). In fact, most of EWS are equipped with similar sensors such camera, water level gauge, geophone, and AWS. Therefore, in the revised manuscript, the relevant description on the specification of ice collapse would be removed. We added the sentence to address the challenges of EWS installation and maintenance in high-altitude region. "*However, the installation and maintenance of EWS in sparsely populated regions generally faces many challenges such as the instrument transport and logistics in high altitude mountainous areas, the harsh extreme weather conditions, the power supply and data transmission in cloudy and rugged regions, the reliability and compatibility of different sensors, and the sustainable funding.*".

The reported successful early warning was also cited (Massey et al., 2010).

Massey, C. I., Manville, V., Hancox, G. H., Keys, H. J., Lawrence, C., and McSaveney, M.: Out-burst flood (lahar) triggered by retrogressive landsliding, 18 March 2007 at Mt Ruapehu, New Zealand—a successful early warning, Landslides, 7, 303-315, 2010.

Section 2 "Study region" and Section 3 "Historical ice-rock collapse around ......" could be merged as Section 2. Study region and historical ice-rock collapses.

Reply: Both reviewers have pointed this out. In the revised manuscript, we have moved it to the study area section. The title of Section 2 has been changed as "Study region and historical ice-rock collapses".

Figure 7 and the supplementary, the seismic waveform of two abnormal waveform (showed by green arrows) need to be displayed in the supplementary. Are these waveforms typically different with the normal waveform introduced by ice-rock avalanche? Maybe the small-scale rock avalanches without debris flow formation.

Reply: Thank you for your suggestions. We have provided the seismic waveform of these two abnormal waveforms in the supplementary(Figure S9). The waveforms on 2 June and 1 October 2022 are completely different from the typical avalanche-induced waveform. Although the amplitude of waveform is above 20, the duration and characteristics of waveform is different. The typical avalanche-induced waveform usually lasts several minutes and displays the gradually decreasing amplitude with the collapses. However, the two abnormal waveforms are very short, indicating the possible waveform noise. This is the reason why no debris-flow could be verified in the outlet of the Sedongpu valley.

[Figure]

**Figure S9**: The two abnormal waveforms occurred on 2 June 2022 (a) and 1 October 2022 (b) and the collapse-induced waveforms occurred on 11 August 2022 (c), showing the different waveform and duration.

Line 276-278, the author claimed that both the maximum amplitude and the duration of seismic waveform are useful information for early warning the ice-rock collapse. Did the EWS incorporate this information in the system? In Line 174-175, the warning system seem to only consider the thresholds as the only warning information. More explains should be added for introduce the automatic warning system in the manuscript.

Reply: In the current stage of the warning system, we used the threshold of the waveform amplitude as the automatic warning path. If the amplitude is over 20, the warning message is sent to the two experts in the office of the Second Tibetan Plateau Scientific Expedition and Research Program. The experts will check the multi-parameters including real-time photos, videos, water level and waveform characters to determine the status of the glacier and river and inform the local government. We have added this information at the beginning of section 4.

Line 295-Line 300. The author provided the values of anomalous glacier thickening on three risking glaciers in the ablation zone. The boundary of this ablation zone should be delineated in Figure 9.

Reply: Following your suggestions, we have added the boundary of the ablation zones in the revised Figure 9.

[Figure]

**Figure 9.** Spatial distribution of mean surface elevation change (m/y) during 2010-2020 (Hugonnet et al., 2021) and annual surface velocity (m/day) in 2017 (Millan et al., 2022),

and the locations of the three glaciers with abnormal surface thickening (the red rectangles) and the tower of Pai.

The EWS in this manuscript did not include the GNSS for monitoring glacier displacement, which is the direct indicator of glacier abnormal change and the popular way for disaster monitoring (landslide and rock avalanche). Could the GNSS be used on the glacier surface, in particular for the risking glaciers such as the Zelongnong Glacier and the other glacier RGI60-13.01430 for early warning their abnormal dynamic changes?

Reply: Installing high-resolution GNSS on the glacier surface is a good way to monitor the abnormal glacier dynamics with sufficient temporal resolution. However, for the Sedongpu Glacier and RGI60-13.01430, it is very difficult to install and maintain GNSS due to the logical problem. However, for the debris-covered Zelongnong Glacier, GNSS is the best way to monitor its dynamics. We will plan to install GNSS on this glacier in the near future. We have included this monitoring method in the revised manuscript. "*Enhanced monitoring, particular the surface movement monitoring by using Global Navigation Satellite System (GNSS), is critical for the accessible Zelongnong Glacier, which is very close to the town of Pai with a population of more than 3000, and for the proposed mega hydropower project nearby.*"

In section 6.2, the author stated that all-weather avalanche radar for real-time monitoring. Do the author know the similar instruments available for EWS?

Reply: The GEOPREVENT in Switzerland has developed the Avalanche Radar for real-time monitoring with a range of up to 5 km. Detailed information is available in https://www.geopraevent.ch/technologies/avalanche-radar/?lang=en

**Minor comments:**

Line 130. The video of the disaster process is retrieved remotely.

Reply: we have added "by commands" at the end.

Figure 1. Yarlung Tsangpo, Figure 4. Yarlungzangbo River. I suggest consistent use of Yarlung Tsangpo throughout.

Reply: Thanks. We have used the Yarlung Tsangpo in the whole manuscript.

The quality of Figure 5 should be improved to be more than 300Mdpi. The red arrow was not displayed fully in Figure 5a. and the major tick lines in X and Y axis should be added.

Reply: Thanks. We have improved the quality of Figure 5 and added/revised the relevant problems.

[Figure]

**Figure 5.** The recorded waveform of seven collapses on 11 August 2022 (a) and the water level rise by small-scale debris flow and precipitation condition (b), and the photos taken before and after the occurrence of fresh debris flow into the Yarlung Tsangpo River (c,d), and video screenshots of two rock collapses at 16:24 and 16:57 on 11 August 2022.

Figure 9a, the units is m or m/year?

Reply: We have changed it as m/y. Thanks.

**Response to Reviewer 2**

**General comments:**

I had the chance to review the paper 'Early warning system for ice collapses and river blockages in the Sedongpu Valley, southeastern Tibetan Plateau'. The paper analyses the performance of a three sets early warning system in Sedongpu valley to detect ice rock collapse and alert the communities. Such an analysis is very useful for the emergency and scientific community and while the novelty of the approach is moderate, I recommend this paper for NHESS with major revisions.

Overall, the paper is very interesting, and the data well presented. The strengths of this paper are the EWSs setups which fed the paper and made the analysis relevant. The efficiency of EWS 1, 2 and 3 data is well described. Evidence on the waveform characteristics of a flow is very informative as well as the dilution of the flow when rocks and ice are mixed. The charts are clear, although I have made some suggestions below. While this paper will find readers from the scientific and the emergency stakeholders, I would recommend to further improve the scientific analysis and take full advantage of this EWSs' network. The EWSs network introduced here offer a great source of dataset which could be further exploited in this paper. Particularly, EWS1 meteorological records should have been analysed to understand the role of weather parameters on ice-rock detachment (Temperature? Precipitation? Radiation?). EWS1 was established in May 2022, therefore it would be very useful to look at the data records prior to Ice-rock collapses and river blockage on 14 May 2022 and to the repeated ice-rock collapses on 11 August 2022. If there are any anomalies prior to the events, those must be mentioned in the paper to better understand the weather conditions which trigger such hazards. This is the main, if not the only, gap of this paper. Please, see some of my comments in details below.

Reply: Thank you for your positive assessment of our work and your constructive comments, which will help us to improve the paper considerably. Following your suggestion, we have re-examined the meteorological data to analyze the relationship between the occurrence of ice-rock collapses and the possible weather conditions that could trigger the hazards in the Sedongpu valley. The following new Figure 8 shows the occurrence of ice-rock collapse and the time series of meteorological factors such as air temperature, precipitation, wind speed and incoming shortwave radiation. Table S1 also lists the detailed values of meteorological factors during the occurrence of ice-rock collapse in 2022.

Both Figure 8 and Table S1 show that the ice-rock collapses could occur under different meteorological combinations. For example, the weather condition of ice-rock collapse on 14 May 2022 was characterized by relative colder (8.3 °C), heavy rainfall (22.6 mm), weaker wind speed (0.3 m/s) and lower solar radiation (83.9 W/m²). In contrast, the weather condition of ice-rock collapse on 11 August 2022 was warmer (15.4 °C), less precipitation (0.4 mm), moderate wind speed (0.7 m/s) and sunny weather (321.1 W/m²). In addition,

Figure 8 also showed no clear anomalies prior to the ice-rock events (e.g. the extreme high air temperature or heavy rainfall). This suggests that there were no immediate meteorological triggers for the ice-rock collapses and the associated debris flows in 2022 at the Sedongpu Valley. The main triggers of ice-rock collapse are worth investigating from the instability of temperate glaciers, rock properties and associated freeze-thaw weathering, pre-event seismic activity, the cumulative role of climate change and other possible factors. It is unfortunate, however, that there are few data in the high-altitude collapse regimes in the Sedongpu Valley to clarify this important issue.

Following the review comments, we have added the following Figure 8 in Section 5.1 (Table S1 in the Supplement) to discuss the possibility between the occurrence of ice-rock collapse and weather conditions. Such figure is helpful for the reader to understand the meteorological conditions in the Sedongpu valley and to answer the possible confusion about the relationship between extreme weather conditions and ice-rock collapses.

[Figure]

**Figure 8.** The occurrence of repeated ice-rock collapses (red stars) and the time-series of meteorological factors including air temperature (a), rainfall (b), wind speed (c) from EWS1 and the incoming shortwave radiation from EWS3 (d).

**Table S1**. The daily mean values of meteorological factors during the occurrence of ice-rock collapses in 2022

| Collapse Time | Air temperature (°C) | Rainfall (mm) | Wind speed (m/s) | Solar radiation (W/m$^2$) |
|---|---|---|---|---|
| 2022/5/14 | 8.3 | 22.6 | 0.3 | 83.9 |
| 2022/7/1 | 13.7 | 0.0 | 0.9 | 314.0 |
| 2022/8/7 | 15.8 | 2.8 | 0.8 | 301.0 |
| 2022/8/11 | 15.4 | 0.4 | 0.7 | 321.1 |
| 2022/9/9 | 10.5 | 6.4 | 0.5 | 184.3 |
| 2022/9/19 | 13.6 | 0.0 | 0.7 | 216.6 |
| 2022/9/24 | 9.6 | 9.4 | 0.3 | 107.3 |
| 2022/10/20 | 4.6 | 0.0 | 0.9 | 166.8 |
| 2022/11/2 | 4.0 | 0.0 | 1.3 | 146.4 |

**Scientific Significance: Fair.** All statements are supported by clear evidence which contributes to the understanding of these natural hazards. However, weather parameters should be further exploited to explain the weather conditions of such hazards in Sedongpu.

Reply: As the reply in above, we have added the new Section 5.1 in the revised manuscript to discuss the linkage between the occurrence of ice-rock collapses and the abnormal weather condition in the Sedongpu valley.

**Scientific Quality: Good.** The paper is based on exclusive observations of two remote rivers. Some references could be added to the paper to better place this paper in a global context.

Reply: We have the following references in the revised manuscript.

Massey, C. I., Manville, V., Hancox, G. H., Keys, H. J., Lawrence, C., and McSaveney, M.: Out-burst flood (lahar) triggered by retrogressive landsliding, 18 March 2007 at Mt Ruapehu, New Zealand—a successful early warning, Landslides, 7, 303-315, 2010.

Haemmig, C., Huss, M., Keusen, H., Hess, J., Wegmüller, U., Ao, Z., and Kulubayi, W.: Hazard assessment of glacial lake outburst floods from Kyagar glacier, Karakoram mountains, China, Annals of Glaciology, 55, 34-44, 2014.

Tiwari, A., Sain, K., Kumar, A., Tiwari, J., Paul, A., Kumar, N., Haldar, C., Kumar, S., and Pandey, C. P.: Potential seismic precursors and surficial dynamics of a deadly Himalayan disaster: an early warning approach, Scientific reports, 12, 3733, 2022.

Stähli, M., Sättele, M., Huggel, C., McArdell, B. W., Lehmann, P., Van Herwijnen, A., Berne, A., Schleiss, M., Ferrari, A., Kos, A., Or, D., and Springman, S. M.: Monitoring and prediction in early warning systems for rapid mass movements, Natural Hazards and Earth System Sciences, 15, 905–917, 2015.

Cook, K. L., Rekapalli, R., Dietze, M., Pilz, M., Cesca, S., Rao, N. P., Srinagesh, D., Paul, H., Metz, M., and Mandal, P.: Detection and potential early warning of catastrophic flow events with regional seismic networks, Science, 374, 87-92, 2021.

**Presentation Quality: Good.** The data are well described and clear. Again, the weather conditions of these hazards are too concise here and should be further explained.

Reply: As the reply in above, we have added the relevant Figure 8, Table S1 in the revised manuscript to discuss the linkage between the occurrence of ice-rock collapses and the abnormal weather condition in the Sedongpu valley.

Line 27: can you provide a figure of the 'rate of air temperature warming'?

Reply: We have added the warming rate of 0.42 °C per decade, which was reported by Yao et al (2022) and the relevant referenced papers.

Line 46: 'Given the short duration of glacier collapse, it is difficult to provide timely warnings of glacier catastrophes and assess their impacts', add 'using remote sensing only' at the end of this sentence, to emphasize your point about remote sensing's limitations.

Reply: Following your suggestion, we have added it.

Line 49-51: provide more explanations about the difficulties faced with those EWSs. The reader needs to understand the limitations of EWSs, and how you are going to address these issues in your paper.

Reply: We completely agree with your comments. In the revised manuscript, we have added one sentence to address the difficulties faced with the ground-based EWS in the mountainous regimes. The following sentence will be added in the paper.

"*Ground-based early warning systems (EWS) provide a real-time monitoring dataset for warning against catastrophes. However, the installation and maintenance of EWS in sparsely populated regions generally faces many challenges such as the instrument transport and logistics in high altitude mountainous areas, the harsh extreme weather conditions, the power supply and data transmission in cloudy and rugged regions, the reliability and compatibility of different sensors, and the sustainable funding.*"

In the objective paragraph line 52 to 59, I would suggest being more specific on your objective, rather than mentioning the EWS only. For instance, on what real-time physical measurements the real-time warnings signals are based. Water level and other parameters

such as soil moisture or precipitation? We find information about this only in caption of Figure 1.

Reply: Thanks for pointing out this issue. In the revised manuscript, we have re-organized the objective to address the performance of different signals for warning the occurrence and process of different types of ice-rock collapses.

*"The aim of this study is to introduce the structure of three EWSs installed near/inside the Sedongpu Valley, to analyse the performance of different monitoring signals (e.g. water level, geophone waveform, meteorological variables, optical/thermal images) on warning the occurrence and process of different types of ice-rock collapses (ice-rock mixed or rock-dominated events) and finally to discuss the possible monitoring priority and challenge for ice-rock disasters on the Tibetan Plateau."*

How your EWS is more transferrable to other regions than other EWS, as stated line 58-59. We need to have a clue of the real novelty of your approach at this point of the paper. All EWS use real-time signals.

Reply: Thank for this comment. You are right, all EWS use the real-time signals. In the revised manuscript, the pioneering work was not mentioned any more. In the Section 5.3, one paragraph has been added to address the challenges of our EWS and other EWS.

Line 57: 'different types of ice-rock collapse', define them here in few words.

Reply: we have list two types of ice-rock collapse in the revised manuscript as "*ice-rock mixed or rock-dominated events*"

Line 54: I would, conventionally, keep 'Second Tibetan Plateau Scientific Expedition and Research Program' mentioned line 54 to the acknowledgement section.

Reply: Following your suggestion, we have moved it to the acknowledgement section.

Line 67: "There were two peaks over 7000 m above sea level (m asl): Mt. Namcha Barwa (7782 m asl) and Mt.Gyala Peri (7294 m asl)." Please change 'were' into 'are', those peaks still exist.

Reply: Following your suggestion, we have revised it in the revised manuscript.

Line 70: when the summer monsoon is mentioned, I suggest to provide the annual precipitation mean of Sedongpu Valley, and its seasonal distribution (% per season). Moreover, it would be useful to know the share of liquid and solid precipitation. Heavy precipitation, and therefore saturated soil moisture can have an impact on ice-rock collapses.

Reply: Following the reviewer's suggestion, we provide the annual precipitation and seasonal distribution of Medog station in which is about 60km from the Sedongpu valley

in the revised manuscript. In addition, there are less information on the solid and liquid precipitation in both stations and AWS in the Sedongpu valley. However, we provide both precipitation and air temperature variation in the Sedongpu Valley during the period from April to December. From the Figure 8, the mean daily air temperature was generally higher than zero degree. Therefore, we assumed that most of precipitation fall as rainfall at the elevation of 3300m near EWS1.

"*Total precipitation in Medog, located about 60 km from the Sedonpu Valley, was more than 1200 mm during 2019-2020, with 56.6% falling in June-September and 32.4% in the spring season (March-May) (Li et al., 2022b)*"

Section 3 on the history of ice rocks collapses is very informative and interesting. I suggest however to move it to the study area section (section 2) as it is not part of the EWS' results but rather a review of previous studies on this region. Moreover, information about the consequences (or even the absence of severe destructions) of the cited debris flows on villages in the lower part are needed here, in order to better understand the challenges of this valley. What happened to the villages downstream or the bridge of Gyala after the 22 October 2017 for example, once the water was released?

Reply: Both reviewers have pointed out this issue. In the revised manuscript, we have moved it to the study area section. The title of Section 2 has been changed as "Study region and historical ice-rock collapses". And following your suggestion, we have added some text to address the consequences of river blockages in the revised manuscript.

For the 2018 and 2021 blockage, we have added the flowing text "*This event block the Yarlung Tsangpo River for ~60 h and the river level increased about 75m above the original level, which damaged two upstream bridges and inundated dozens kilometres of roads and power supply facilities and forced the evacuation of at least 6000 local resident (Chen et al., 2020). The blocked dam was overtopped on 19 October, with the peak breaching flow as large as 32000 m$^3$/s and damaged the downstream hydropower station. On 22 March 2021, massive ice-rock collapses totalling 50 Mm$^3$ occurred in the Sedongpu Valley, producing a mobile debris flow that temporarily blocked the Yarlung Tsangpo River, leading to the inundation of road to Gyala village (Zhao et al., 2022).*"

For the Zelongnong valley, we have added the flowing text "*River blockages have been reported to have occurred in 1950, 1968 and 1984 and the glacier collapse in 1950 engulfed the village of Zhibai and lead to the death of to 97 villagers (Zhang, 1992). In 2020, a total of 1.14 Mm$^3$ of ice-debris mixture produced a high-speed debris flow and partially blocked the Yarlung Tsangpo River and damaged the Zhibai Bridge (Peng et al., 2022)*"

Line 115: 'Since 2019, the EWSs have been built around the Sedongpu Valley.' '*Along'* the valley would better depict the distribution of EWSs based on Fig 1.

Reply: Done

Line 136-137: it would be helpful for the reader to have a brief explanation about the recurrent timing of the small-scale collapses at midnight. Is it due to accumulated solar energy during the day, frost-thaw, or any other factor?

Reply: During the installation of EWS1 in April 2022, we observed several small snow-ice collapses during the day. It is a pity, however, that the monitoring system, including the geophone, was not completed. At midnight the geophone system was running continuously and fortunately several collapses occurred and were clearly reflected in the geophone waveform. Therefore, due to the large amount of useless waveform data and the limited capability of satellite transmission, if any XYZ vector was greater than three (the maximum amplitude of the collapse in the midnight), the 200 seconds of waveform data before and after the threshold were automatically transmitted to the server. We have aded the relevant description in the revised manuscript.

"*During the installation of EWS1, we witnessed several small-scale snow-ice collapses in daytime and several small-scale collapses occurred at midnight were recorded by the geophone. The corresponding amplitude of the three-component waveform was generally greater than three when the collapse occurred. Therefore, if any XYZ vector was greater than three, the 200-second waveform data before and after the threshold were transmitted automatically to the server*"

I suppose the meteorological parameters were recorded on an hourly basis. Please precise it in the text.

Reply: Meteorological parameters were stored at 30 min. In the revised manuscript, we have added this information.

"*With regard to possible triggers of extreme weather conditions for ice-rock collapses and the lack of meteorological data in the Sedongpu Valley, meteorological variables were recorded every 30min using the Campbell datalogger and were transmitted to the server*"

Line 184: what sensor was used and what resolution?

Reply: we have added the details of sensor e.g. pixels for thermal/optical and resolution of water level sensor in the revised Table 1.

Figure 3a, 5a, precise the units of the amplitude.

Reply: The units of the amplitude (V/m/s) have added in the revised manuscript.

Line 212-213: 'which agrees well with the delay between the observed ice collapse in October 2019 and its delayed debris flow (Zhao et al., 2022)', please provide a value (hours) of the time of response between the collapse and the actual water level rise on that day.

Reply: According to the geophone and water level records, the time of rive blockage by the debris-flow was about 20 hours later than the first collapse. We have added this information in the revised manuscript.

*"The delayed formation of debris flows after ice-rock collapses (about 20 hours later than the first collapse) suggests that it takes some time for the collapsed ice to melt, which agrees well with the delay between the observed ice collapse in October 2019 and its delayed debris flow (Zhao et al., 2022)."*

Figure 5b, the photo is not visible and deserves a better resolution. I suggest increasing its size below Fig5.

Reply: Following your suggestion, the photo was enlarged and the photo before the occurrence of debris flow was also provided to highlight the fresh debris flow.

[Figure]

**Figure 5.** The recorded waveform of seven collapses on 11 August 2022 (a) and the water level rise by small-scale debris flow and precipitation condition (b), and the photos taken before and after the occurrence of fresh debris flow into the Yarlung Tsangpo River (c,d), and video screenshots of two rock collapses at 16:24 and 16:57 on 11 August 2022.

Line 226: 'EMS1', EWS1?

Reply: We have changed it as EWS1.

In Figure 8 (very informative), it would be better to have the same amplitude axis to better visualize the difference between collapse and seismic waveforms.

Reply: Following your suggestion, we have unified the amplitude of both X (a total of 20 mins) and Y (ranging from -40 to +40 for ice-rock collapse and -80 to +80 for earthquake) axis to show the difference between ice/rock collapse and seismic waveforms in the duration and waveform morphology. Please see the revised Figure as followings:

[Figure]

**Figure 7.** Comparison of the waveforms generated by ice-rock collapses (a-i) and by the earthquake on 10 November 2022 (j). Note the different y-axis limits.

In Figure 8, add the Hugonnet and Millan references in the caption.

Reply: We did it.

In the discussion, you may want to go through Tiwari et al., 2022 and compare your approaches using amplitudes. (Potential seismic precursors and surficial dynamics of a deadly Himalayan disaster: an early warning; https://doi.org/10.1038/s41598-022-07491-y). It's worth adding this reference in the discussion section. See also Cook et al., 2021: 'Detection and potential early warning of catastrophic flow events with regional seismic networks', DOI: 10.1126/science.abj122

Reply: Thank you very much for this comment. We have cited these two important references in the revised manuscript. Detailed retrospective analysis of the waveform

showed that some possible precursors were present before the collapses. However, regarding the possible noise is concerned, there is no solid evidence that such signals could be used as the pre-warning way. In section 5.3, we have discussed the challenges and priority of seismic warning and more relevant references have been cited in the revised manuscript.

"*Previous studies have shown that seismic observations are an important way to understand the dynamic process of different catastrophic disasters such as ice-rock collapses (Bai et al., 2023; Cook et al., 2022; Tiwari et al., 2022), glacier lake outburst floods (Cook et al., 2018; Maurer et al., 2020), rock landslides (Le Breton et al., 2021), and have important potential for detecting the precursory signals before the event as an early warning by the distant seismic stations up to 100 km from the disaster (Cook et al., 2022). Seismic geophones are widely used in EWS around the world and are one of the main early warning tools (Massey et al., 2010). Although catastrophic events can be detected in seismic data, the general limits and controls on detection and location remain challenged. Seismic energy generated in surficial catastrophic activities is easily saturated or attenuated with distance (West et al., 2010). For example, the geophone at EWS1 inside the Sedongpu Valley provided a good warning due to its proximity to the source of the collapses (Fig. 1). In contrast, the geophone at EWS2 near the Yarlung Tsangpo River was significantly influenced by river-generated noise and was less sensitive to the small-scale collapses in 2022 due to its distance from the source regimes. The installation of seismic stations for event detection is therefore a function of the desired minimum detectable event size. The deployment of the optimized seismic stations is crucial to obtain the high-quality signal with less noise ratio in the rugged mountainous regions (Cook et al., 2022). And the reliability of seismic warning should be further confirmed and validated by using multi-parameter observations (e.g. remote sensing data, river gauges, photographs and videos). The concurrent multi-parameter observations (water level, thermal/optical images, and video) provided the opportunity to validate the occurrence of the above-mentioned nine repeated ice-rock collapses in 2022 and the reliability of the seismic threshold for early warning in the Sedongpu valley. The possible false warns can also be excluded by analysing the duration and characteristics of the waveforms and the corresponding in-situ observations. However, it should be noted that this local threshold (amplitude greater than 20) is empirically appropriate for EWS1. In fact, each phase of the disaster chain has a different seismic signature and a different detection limit at far-field stations (Massey et al., 2010). The local threshold for seismic warning should be determined by historical records of ice-rock collapse events and validated by the field multi-parameter observations in other hazard regions. In addition, the detection of precursory signals prior to the ice-rock collapse for early warning is very promising but remains a major challenge (Cook et al., 2022; Tiwar et al., 2022). Automated processing by using innovative technologies such as deep learning methods would be helpful for the effective future warning of catastrophic ice-rock collapse (Tiwar et al., 2022).*"

**Response to Reviewer 3**

**General comments:**

The authors present and impressive EWS setup in the Namche Barwa massif, an area that has been subject to multiple hazard cascades in recent years and decades. They present a combination of sensors at different locations and based on a limited number of months of observations are able to track movements and show the potential of these setups for early warning. Especially useful, as has been shown in other sites, is a geophone that is able to pick up mass movements of different magnitude well before they become the actual risk when damming the main river channel.

The study is well presented and the setup and the insights present a valuable contribution to NHESS. Before recommending it for publication the authors however need to embed their findings into the state of knowledge on EWS in the region, especially in light of their geophone/wavelet analysis angle. Other studies exist and those need to be brought into context to make this not just a description of a successful first test of the setup but a contribution that will be helpful for the scientific community. NHESS requires this to be a novel contribution, and at the moment I miss a discussion of the added value (apart from of course the value the specific EWS has for local stakeholders). While this requires some more work on the Discussion, I believe that seeing the quality of the manuscript up to this stage this should be well possible to do.

Reply: We are grateful to Dr. Jakob Steiner for the detailed and insightful comments and suggestions. We have substantially restructured the Discussion section to discuss the relationship between the occurrence of ice-rock collapses and meteorological factors, and to address the multi-parameter observations for early warning. The relevant previous published studies have been referenced and the possible challenges of the EWS have been discussed following your suggestions. Please see the detailed responses to the main and specific comments below.

**Main comments**

L153: What are the 'various variables'? There is also no discussion on how the meteo data is useful or used. Are they required at all? What role do they play in the EWS chain of communication/analysis?

Reply: The meteorological variables in the 10m integrated observation tower of the destroyed EWS2 includes wind speed/direction, air temperature, relative humidity, four components of radiation, rainfall, atmospheric pressure. We have added this information in the revised manuscript.

"*a 10 m integrated observation tower, equipped with time-lapse optical and thermal cameras and various meteorological variables (wind speed/direction, air temperature, relative humidity, four components of radiation, rainfall, atmospheric pressure), was installed 150 m above the valley floor at the valley outlet in October 2019*"

In fact, there are no previous meteorological records in this unpopulated Sedongpu valley. Therefore, these in-situ AWS measurements were designed to investigate the possible meteorological triggers (e.g. extreme high air temperature/heavy rainfall) for the ice-rock collapses and debris mass flows, and also to provide the local meteorological background for further study of the mechanism and process of ice-rock collapses based on the glacier thermo-dynamic models. In the revised manuscript, we have added a new section 5.1 to discuss the relationship between the occurrence of repeated ice-rock collapses and the time series of meteorological factors. The following new figure will be included in the revised manuscript to help the reader understand that there are no immediate meteorological triggers for the ice-rock collapses. Please see the new Section 5.1.

[Figure]

**Figure 8**. The occurrence of repeated ice-rock collapses (red stars) and the time-series of meteorological factors including air temperature (a), rainfall (b), wind speed (c) from EWS1 and the incoming shortwave radiation from EWS3 (d).

Figure 4: It is very hard for me to make out changes on these images. In a/b can you indicate flow direction, what the deposit and what water is? In c to f I can not really see change

between the two images. Can you indicate what change the reader should perceive? It would also be important in the text to clarify how these images are used in early warning – to validate? Is there an algorithm that traks changes on the images as was the case at Kyagar? Stick to one writing of Yarlung Tsangpo/Yarlungzangbo

Reply: Following your suggestion, the flow direction of the Yalung Tsangpo River (white colour in the thermal image) has been added to the revised figure. Due to the contrasting surface temperature, the colour of the cold debris flow was dark in the thermal image. We have highlighted the fresh debris in the image.

And we have enlarged the source region to show its changes by comparing two thermal images. Please see the following new Figure 4. The collapse source on the ridge of Mt. Gyala Peri was also manually delimited by comparing Figure 4 e,f.

[Figure]

**Figure 4.** The debris-induced blockage of the Yarlung Tsangpo River (a, b) and the optical and thermal images showing the topographic conditions at the targeted monitoring of the collapsed area (c, d) with the zoomed region (red dashed rectangle) shown in e-f, and the change of ridgeline before and after the repeated ice-rock collapses on 14 May 2022 and the extent of the collapse source are outlined in transparent red.

Due to the influence of cloudy and rainy weather in the Sedongpu Valley, both optical and infrared photographs at EWS1 hardly capture the cloud-free photographs during the monsoon season and are therefore not applicable for use as real-time warning indicators. The selected pre- and post-collapse photographs are generally used to determine the location and extent of collapse. In the revised manuscript, we have added the relevant text to clarify the purpose of the photographs in section 3.1.

"*Both optical and infrared photographs are often affected by the heavy cloud cover and rainfall in this high-altitude region during the monsoon season, making them unsuitable as the real-time warning indicators. This targeted monitoring was designed to identify the location and magnitude of the recurrent collapses by comparing the pre- and post-event photographs.*"

In contrast, EWS2 near the outlet of the Sedongpu valley tends to have the highest quality imagery due to less cloud cover and the shorter distance to the blockage. Therefore, the repeated comparison of optical/infrared images is helpful in determining the time and magnitude of the river blockage. In fact, we are now trying to use the deep learning method to aromatically outline the extent of debris fans and the river area. Such information would be an important automatic way to complement the current real-time early warning system.

We have unified the term as Yarlung Tsangpo river.

Figure 6: As in Figure 4 I find it hard to see anything in c to f. Would it be possible to zoom in for the purpose of the manuscript? You need to tell the reader how these images are useful for interpretation.

Reply: We acknowledged that it is very difficult to see the change for Figure 4c,f without a dynamic comparison (such as GIF). These figures were therefore moved to the supplementary material (Figure S1). Both Figure 5 and Figure 6 have been combined to show the river blockage and the ice-rock avalanche process (please see the following new figure).

[Figure]

**Figure 5.** The recorded waveform of seven collapses on 11 August 2022 (a) and the water level rise by small-scale debris flow and precipitation condition (b), and the photos taken before and after the occurrence of fresh debris flow into the Yarlung Tsangpo River (c,d), and video screenshots of two rock collapses at 16:24 and 16:57 on 11 August 2022.

L253: Here and then again the Discussion/Conclusion I would expect some critical reflection on potential pitfalls. Why for example did EWS2 not send a warning when the debris flow mass must have reached the location before midnight? Or did it only reach the location that late? You defined a threshold for amplitude based on a few observations, how is this robust? What about false positives or false negatives? The amplitude of the first August 2022 event was larger than the second but you say the second event was larger in volume. What about the smaller events well below 20, would your system then not record them at all and send no potential warning? How does expert judgment of these data and images work, who interprets it and who takes final responsibility for the judgement made? You also introduce a 'three level warning system' based on different amplitudes (L264) but then never explain where this comes from or how it is used. This needs to be clarified.

Reply: Thank you very much for these insightful comments. In the revised manuscript, a new discussion has been added in Section 5.3 to address the above issues. We have discussed the possible challenges and pitfalls in EWS, including your concerns about why EWS2 does not send a warning, the robustness of the warning threshold, and the possible

exclusion of false warnings. The challenges and priority of seismic warning were also discussed.

Firstly, seismic energy generated in surficial catastrophic activities is easily saturated or attenuated with distance. The geophone at EWS1 inside the Sedongpu Valley performed well due to its proximity to the source of the collapse. In contrast, the geophone at EWS2 near the Yarlung Tsangpo River was influenced by river-generated noise and was less sensitive to the small-scale collapse events in 2022 due to its distance from the source regime. However, the geophone at EWS2 would also play the warning role in the event of catastrophic mass flow, as occurred on 22 March 2021 (Zhao et al., 2021). In the revised manuscript, we will address that the installation of seismic stations for event detection is a function of the desired minimum detectable event size and the deployment of the optimized seismic stations is critical important to obtain the signal with high-quality signal-to- noise ratio in the rugged mountainous regions.

Regarding the robustness of the amplitude threshold at EWS1, a total of 9 repeated ice-rock collapses and the associated debris flows in 2022 provided an opportunity to ensure the threshold reliability of the seismic warning (amplitude greater than 20) in the Sedongpu Valley. Each warning event was confirmed by using multi-parameter observations including the photographs, videos, water level. The geophone recorded all signals with amplitude greater than 3. Based on the photographs and videos at EWS1, there were few debris flows occurring when the waveform amplitude at EWS1 was less than 20. In contrast, when the waveform amplitude was greater than 20, the occurrence of debris flows was confirmed at the Sedongpu Valley outlet. There were still two false warnings occurred in June and September. However, the detailed analysis of the waveform can easily exclude such false warnings. The duration and characteristics of the waveform are different, as shown in the following figure (Figure S9). The typical collapse-induced waveform usually lasts several minutes and displays a gradually decreasing amplitude (Figure S9 c). However, the two abnormal waveforms are very short, indicating the possible waveform noise and can be easily excluded by the warning system (Figure S9 ab). However, it should be noted that this local threshold (amplitude above 20) is only suitable for EWS1. Each phase of the disaster chain has a different seismic signature and a different limit of detection at far-field stations. It is recommended that the local thresholds of seismic warning in other hazard regions should be determined by corresponding historical records and the field multi-parameter observations. The above discussion has been added in section 5.3 as followings:

"*Previous studies have shown that seismic observations are an important way to understand the dynamic process of different catastrophic disasters such as ice-rock collapses (Bai et al., 2023; Cook et al., 2022; Tiwari et al., 2022), glacier lake outburst floods (Cook et al., 2018; Maurer et al., 2020), rock landslides (Le Breton et al., 2021), and have important potential for detecting the precursory signals before the event as an early warning by the distant seismic stations up to 100 km from the disaster (Cook et al., 2022). Seismic geophones are widely used in EWS around the world and are one of the main early warning tools (Massey et al., 2010). Although catastrophic events can be*

*detected in seismic data, the general limits and controls on detection and location remain challenged. Seismic energy generated in surficial catastrophic activities is easily saturated or attenuated with distance (West et al., 2010). For example, the geophone at EWS1 inside the Sedongpu Valley provided a good warning due to its proximity to the source of the collapses (Fig. 1). In contrast, the geophone at EWS2 near the Yarlung Tsangpo River was significantly influenced by river-generated noise and was less sensitive to the small-scale collapses in 2022 due to its distance from the source regimes. The installation of seismic stations for event detection is therefore a function of the desired minimum detectable event size. The deployment of the optimized seismic stations is crucial to obtain the high-quality signal with less noise ratio in the rugged mountainous regions (Cook et al., 2022). And the reliability of seismic warning should be further confirmed and validated by using multi-parameter observations (e.g. remote sensing data, river gauges, photographs and videos). The concurrent multi-parameter observations (water level, thermal/optical images, and video) provided the opportunity to validate the occurrence of the above-mentioned nine repeated ice-rock collapses in 2022 and the reliability of the seismic threshold for early warning in the Sedongpu valley. The possible false warns can also be excluded by analysing the duration and characteristics of the waveforms and the corresponding in-situ observations. However, it should be noted that this local threshold (amplitude greater than 20) is empirically appropriate for EWS1. In fact, each phase of the disaster chain has a different seismic signature and a different detection limit at far-field stations (Massey et al., 2010). The local threshold for seismic warning should be determined by historical records of ice-rock collapse events and validated by the field multi-parameter observations in other hazard regions. In addition, the detection of precursory signals prior to the ice-rock collapse for early warning is very promising but remains a major challenge (Cook et al., 2022; Tiwar et al., 2022). Automated processing by using innovative technologies such as deep learning methods would be helpful for the effective future warning of catastrophic ice-rock collapse (Tiwar et al., 2022)."*

[Figure]

**Figure S9**: The two abnormal waveforms occurred on 2 June 2022 (a) and 1 October 2022 (b) and the collapse-induced waveforms occurred on 11 August 2022 (c), showing the different waveform and duration.

The amplitude of the first August 2022 event was larger than that of the second. There is in incorrect expression in the manuscript. We will correct the expression in the revised manuscript.

Whenever any geophone waveform or water level exceeded the specified thresholds, the warning signals were automatically sent to the mobile phones (message and Wetchat) of two responsible experts at the office of the Second Tibetan Plateau Scientific Expedition and Research Program. Subsequently, the observers checked the multi-parameters including the real-time photos, videos, water level and the waveform characters to determine the glacier (EWS1) and river status (EWS2) and informed the local government. The official Disaster Emergency Management Department is responsible to lunch the alarm initiating evacuation. We have added this information in the beginning of Section 4 as followings:

*"The warning signals were automatically sent to the mobile phones of two responsible experts at STPSER whenever any geophone waveform or water level exceeded the specified thresholds. Subsequently, the experts checked the multi-parameters including the real-time photos, videos, water level and the geophone waveform characters to determine the glacier and river status and informed the local government."*

In terms of the three-level warning, in fact, the geophone amplitude of 20 was indeed confirmed as the first-level warning threshold. As no catastrophic events occurred in 2022, the EWS in this study temporally defined twice (40) and three times (60) of the first-level threshold as the second and third warning thresholds, which will be optimized by future observations. We have added such an explanation in the revised manuscript as followings:

"*Therefore, when the geophone amplitude exceeds 20 in EWS1, there is a high probability that collapse-induced debris flow will occur in the Sedongpu Valley. The geophone amplitude of 20 was selected as the first-level warning threshold. Due to no catastrophic events occurred in 2022, the EWS defined temporally two times (40) and three times (60) of the first-level threshold as the second and third-level warning thresholds, which would be optimized by further observations.*"

In Figure 8 and the text above you very briefly touch upon this topic but I think this needs to be further expanded. What about the noise before the rock collapse, not visible before a rupture, could this be exploited as a prewarning signal?

Reply: Following your suggestion, we have compared in detail the waveforms between nine ice-rock collapses and seismic activity in new figure (Please see the following figure). We clarified that the waveform of a typical ice-rock collapse is completely different from that of an earthquake in terms of duration and waveform morphology. Detailed retrospective analysis of the waveform also showed that some possible precursors were present before the collapses. However, regarding the possible noise is concerned, there is no solid evidence that such signals could be used as the pre-warning way. In fact, the amplitude of the waveform is still the most practical and simple way to warn the occurrence of collapses.

[Figure]

**Figure 7.** Comparison of the waveforms generated by ice-rock collapses (a-i) and by the earthquake on 10 November 2022 (j). Note the different y-axis limits.

L282: The Discussion misses two crucial points:

1. a discussion of the presented approach keeping in mind similar approaches, one of which you mentioned but did not discuss (Maurer et al. 2020) and one that didn't occur at all (Cook et al. 2021). Both these studies have discussed the potential especially of wavelets in the region for similar events and it is crucial that for a scientific publication the previous literature is looked into and put into context for a solid conclusion on a way forward.

Reply: Thank you for pointing this out. Previous studies have shown that seismic waveform is an important way to understand the dynamic process of various catastrophic disasters such as ice-rock collapse, glacier lake outburst flood, rock landslide, and has important potential for detecting the precursor signals before the event as an early warning by the distant seismic stations up to 100 km from the disaster. The EWSs in the world are generally equipped with geophones, which are one of the main instruments. In the revised manuscript, the relevant studies including Cook 2021; Tiwari et al. 2022; Stähli et al. 2015; Massey et al. 2010 Haemming et al. 2014 were cited in Section 5.3.

Massey, C. I., Manville, V., Hancox, G. H., Keys, H. J., Lawrence, C., and McSaveney, M.: Out-burst flood (lahar) triggered by retrogressive landsliding, 18 March 2007 at Mt Ruapehu, New Zealand—a successful early warning, Landslides, 7, 303-315, 2010.

Haemmig, C., Huss, M., Keusen, H., Hess, J., Wegmüller, U., Ao, Z., and Kulubayi, W.: Hazard assessment of glacial lake outburst floods from Kyagar glacier, Karakoram mountains, China, Annals of Glaciology, 55, 34-44, 2014.

Tiwari, A., Sain, K., Kumar, A., Tiwari, J., Paul, A., Kumar, N., Haldar, C., Kumar, S., and Pandey, C. P.: Potential seismic precursors and surficial dynamics of a deadly Himalayan disaster: an early warning approach, Scientific reports, 12, 3733, 2022.

Stähli, M., Sättele, M., Huggel, C., McArdell, B. W., Lehmann, P., Van Herwijnen, A., Berne, A., Schleiss, M., Ferrari, A., Kos, A., Or, D., and Springman, S. M.: Monitoring and prediction in early warning systems for rapid mass movements, Nat. Hazards Earth Syst. Sci., 15, 905–917, https://doi.org/10.5194/nhess-15-905-2015, 2015.

Cook, K. L., Rekapalli, R., Dietze, M., Pilz, M., Cesca, S., Rao, N. P., Srinagesh, D., Paul, H., Metz, M., and Mandal, P.: Detection and potential early warning of catastrophic flow events with regional seismic networks, Science, 374, 87-92, 2021.

2. A discussion of the institutional challenges. The authors are aware of the challenges within China (e.g. Kyagar) but also with other countries of the region (e.g. downstream Nepal) to make such systems work. How is this setup managed locally for success? Are recommendations understood and transported on? What stakeholders need to be included. I realize that this is not the main part of the manuscript and also doesn't need to take up a part in the Results but should surface in the Discussion. Otherwise many of the technical observations often become unsustainable. Wanting a radar is nice, but so expensive that it is hardly ever feasible and you need to chain of command at the end of a signal too to turn such data into actual actionable information.

Reply: Thank you for this insightful comment. We fully agree with your comments. The installation and maintenance of many EWSs generally face institutional challenges and are necessary to gain government support and recognition from local people, especially those with strong religious beliefs. The repeated destruction caused by river blockages improved the knowledge of the local population and the government about the ice-rock collapse and the importance of an early warning system in the Sedongpu valley. As a result, we received strong support from the local government. The official Disaster Emergency Management Department is responsible for triggering the evacuation alarm when the massive ice collapses and associated river blockages are warned and further confirmed by experts and officials. The implementation of similar EWS in other high-risk regions should not only focus on monitoring techniques, but also pay much attention to the cognitive and response capabilities of the government and the population. We have added this discussion in section 5.3.

"*Moreover, the installation and maintenance of the EWS generally faces the institutional challenges and is necessary to gain the support of the local government and the acceptance of the local population, especially those with strong religious beliefs (Huggel et al., 2020; Sufri et al., 2020). The repeated destruction caused by river blockages improved the local population's and government's knowledge of the ice-rock collapses and the importance of an early warning system in the Sedongpu Valley. As a result, we gained strong support from the local government. The official Disaster Emergency Management Department is responsible to lunch the alarm initiating evacuation when the serious river blockages are confirmed. The implementation of similar EWS in other high-risk regions should not only focus on the monitoring techniques, but should also pay close attention to the cognitive and response capabilities of the government and local residents.*"

Radar is expensive and is unlikely to be feasible for all EWS in the Himalayas and Tibetan Plateau. For some key regions related to large infrastructure (e.g. hydropower dams), we think such methods are worthwhile to detect deformation and instability of rock and ice in high altitude glaciated regions. We are now trying to develop the relatively inexpensive radar with Chinese companies. We hope that such radar could

be used to monitor cryospheric disasters such as snow avalanches and ice-rock collapses.

"*Therefore, new weather-independent monitoring instruments (e.g. all-weather avalanche radar) should be considered for real-time monitoring and for accessing the location and volume of ice-rock collapses in the next updated ground-based EWS at some key locations related to the large infrastructure (e.g. mega hydropower stations).*"

Minor comments

L32: remove 'giant', not a relevant descriptor

Reply: We have changed it.

L45: 'before a glacier collapse'

Reply: We have changed it.

L45f: '…data is subject to favourable weather conditions and revisit cycles.'

Reply: We have changed it.

L46: 'a glacier collapse' – also state here what you mean by short, what is the time frame to be expected from known cases.

Reply: The duration was about 5 minutes for 2021 collapse in Sedongpu Valley (Zhao et al., 2022) and was about 2–3 minutes for the 2016 Aru collapse (Kääb et al, 2018). We have added this time duration in the revised manuscript.

L47: 'glacier catastrophes' is a new term you now use while earlier you used 'glacier-related disasters' or just specifically 'glacier collapse'. I would stick to one and only use another if strictly necessary to describe something else.

Reply: Thank for pointing out this issue. We have unified the term as 'glacier collapse'.

L49: remove 'regimes'

Reply: We did it.

L67: 'are' not 'were'

Reply: We changed it.

L128/L151: As you are here describing a setup, details for all sensors are crucial. What kind of thermal and why a thermal camera - refer to the Table where appropriate.

Reply: Thanks for this comment. In the Table 1, we have added the details of sensor e.g. pixels for thermal/optical and resolution of water level sensor in the revised manuscript.

L180: 'is briefly described'

Reply: Done

Figure 3b: Legend missing making clear what is level and what is precipitation

Reply: We have added the legend in the revised manuscript.

[Figure]

**Figure 3.** The recorded geophone waveform at EWS1 due to frequent ice-rock collapses on 14 May 2022 (a), and the rising water level of the Yarlung Tsangpo River due to the blockage on 15 May 2022 along with the hourly precipitation (b)

L262: Spelling is generally 'Medog' in local Tibetan or if you use Chinese placenames 'Motuo'

Reply: We have changed it as Medog.

---

## Author Response (AR2)

**Reply to the Reviewers**

**RC1**

I suggest the manuscript could be accepted after a minor revision.

Reply: We greatly appreciate those insightful comments from Dr. Zhuang Yu. In the acknowledgement, we would like to express our appreciation "We would like to sincerely acknowledge and express our deep appreciation to Dr. Zhang Yu, Dr. Jakob Steiner and the anonymous reviewer for their thorough review of this work."

**RC2**

The main point raised in RC2, which is the investigation of the concurrence of ice-collapses with weather anomalies or extreme events, has been addressed by the authors. This matter has been subject to debate. The authors have successfully demonstrated that there are no significant correlations between the collapse and any weather conditions. They state, "The main triggers of ice-rock collapse are worth investigating from the instability of temperate glaciers, rock properties and associated freeze-thaw weathering, pre-event seismic activity, the cumulative role of climate change and other possible factors." This will be the focus of the team's future research. Figure 8 in the new section 5.1 and Table S1 in the Supplement provide insights into these hazards, contradicting the common belief that heavy rainfalls alone are responsible. Instead, cumulative processes such as freeze-thaw, seismic activity, and soil moisture play a significant role in the mid to long term leading up to these events. Overall, this very interesting and informative article is now more consistent and suitable for publications, with technical corrections only.

Reply: Thank for the anonymous referee's positive assessment of our work and your constructive comments. In the acknowledgement, we would like to express our appreciation "We would like to sincerely acknowledge and express our deep appreciation to Dr. Zhang Yu, Dr. Jakob Steiner and the anonymous reviewer for their thorough review of this work."

Few technical points to correct in the revised manuscript:

• In the objective section, the authors propose the new sentence: ' "The aim of this study is to introduce the structure of three EWSs installed near/inside the Sedongpu Valley, to analyse the performance of different monitoring'. They must choose between near or inside.

Reply: we choose the term of near in the revised manuscript.

• 'supply facilities and forced the evacuation of at least 6000 local resident', please correct residents

Reply: Done

• 'On 22 March 2021, massive ice-rock collapses totaling 50 Mm3 occurred in the Sedongpu' prefer 'a massive 50 Mm3 ice-rock collapses occurred'

Reply: We have revised it according to your suggestion.

**The comments from the editorial support team on 29 June 2023**

Please ensure that the colour schemes used in your maps and charts allow readers with colour vision deficiencies to correctly interpret your findings. Please check your figures using the Coblis – Color Blindness Simulator (https://www.color-blindness.com/coblis-color-blindness-simulator/) and revise the colour schemes accordingly.

Reply: Thanks for your suggestion. We used the Coblis – Color Blindness Simulator to check all Figures. Figure 1 and Figure 6 was therefore redesign for the readers with color vision deficiencies. Please see the new Figures in the revised manuscript.